# DecentralDC: Assessing data contribution under decentralized sharing and exchange blockchain

Wenjun Ke[1,2], Yulin Liu[3]*, Jiahao Wang[1], Zhi Fang[3], Zangbo Chi[3], Yikai Guo[3], Rui Wang[3], Peng Wang[1,2]

1 School of Computer Science and Engineering, Southeast University, Nanjing, China, 2 Key Laboratory of New Generation Artificial Intelligence Technology and Its Interdisciplinary Applications (Southeast University), Ministry of Education, Nanjing, China, 3 Beijing Institute of Computer Technology and Application, Beijing, China

* liuyulin1024@163.com

## Abstract

The issue of data quality has emerged as a critical concern, as low-quality data can impede data sharing, diminish intrinsic value, and result in economic losses. Current research on data quality assessment primarily focuses on four dimensions: intrinsic, contextual, presentational, and accessibility quality, with intrinsic and presentational quality mainly centered on data content, and contextual quality reflecting data usage scenarios. However, existing approaches lack consideration for the behavior of data within specific application scenarios, which encompasses the degree of participation and support of data within a given scenario, offering valuable insights for optimizing resource deployment and business processes. In response, this paper proposes a data contribution assessment method based on maximal sequential patterns of behavior paradigms (DecentralDC). DecentralDC is composed of three steps: (1) mining the maximal sequential patterns of sharing and exchange behavior paradigms; (2) determining the weights of these paradigms; (3) calculating the contribution of sharing and exchange databases combined with data volume. To validate our approach, two sharing and exchange scenarios of different scales are established. The experimental results in two scenarios validate the effectiveness of our method and demonstrate a significant reduction in cumulative regret and regret rate in data pricing due to the introduction of data contribution. Specifically, compared to the most competitive baseline, the improvements of mean average precision in two scenarios are 6% and 8%. The code and simulation scenarios have been open-sourced and are available at https://github.com/seukgcode/DecentralDC.

**Data Availability Statement:** All relevant data for this study are publicly available from the GitHub repository (https://github.com/seukgcode/DecentralDC).

## Introduction

With the rapid development of artificial intelligence, the Internet of Things, and communication technology, enterprises, organizations, and individuals in various industries have accumulated substantial data resources and formed their own data assets [1–3]. DeMedeiros et al. [4]

**Funding:** This work was supported by National Science Foundation of China (Grant Nos.62376057) and the Start-up Research Fund of Southeast University (RF1028623234). The funders had no role in study design, data collection and analysis, decision to publish, or preparation of the manuscript.

**Competing interests:** NO authors have competing interests.

highlights the extensive use of AI and IoT in various sectors, impacting data accumulation and usage. Abiodun et al. [5] discusses the transformative role of IoT across industries, affecting how data resources are accumulated and managed. Tariq et al. [6] provides insights into the rapid development and implications of IoT technology on data generation and security.

These data assets are vital for social production and operational activities, driving technological advancements and industrial upgrades [7, 8]. Kato et al. [9] discusses the critical role of IoT in enhancing industrial and technological processes, aligning with the notion of data assets being crucial for development. Affia et al. [10] highlights the importance of IoT in healthcare, underlining the vital role of data in improving healthcare services. Mazhar et al. [11] emphasizes how IoT, coupled with AI, is crucial for addressing operational challenges in various sectors. Currently, the issue of data quality [12, 13] has gained prominence. Low-quality data can significantly diminish the intrinsic value of data, obstruct data sharing and exchange, and even lead to substantial economic losses. Elouataoui et al. [14] details the adverse impacts of poor data quality, including economic repercussions, and underscores the critical need for high-quality data for effective organizational decision-making. Cai et al. [15] highlight the difficulties in managing data quality due to the variety and volume of big data, which complicates integration and impacts economic outcomes. Merino et al. [16] emphasize how low-quality data in real-time systems can lead to inefficiencies and financial losses, stressing the importance of quality control. Wamba et al. [17] illustrate that poor data quality undermines firm performance by affecting decision-making and operational efficiency. In this light, the integration of blockchain technology, as extensively detailed in references [18–20] offers a formidable solution by enhancing the security and transparency of data transactions. For instance, Koshiry et al. [18] explores blockchain's role in credential verification within the educational sector, ensuring the accuracy and reliability of academic records. Moreover, the knowledge graph and natural language processing technologies discussed in [19] can significantly enhance the management and retrieval of educational resources, proving essential for modern e-learning platforms. Furthermore, the concerns about data security in public networks, highlighted in [20], underscore the necessity of robust security measures, such as those provided by blockchain, to safeguard sensitive data from unauthorized access. Therefore, data quality assessment is critical to data utilization and the full development of data value.

Research on data quality commenced in the 1970s, yielding significant findings to date. Ehrlinger et al. [21] discusses the evolution and ongoing significance of data quality research, highlighting how the field has developed since the 1970s with a focus on both theoretical and practical applications.The initial step in data quality evaluation is defining the dimensions of assessment. In accordance with scholarly investigations [22–28] (Fig 1), the prevailing focus of attention rests upon four principal categories of quality dimensions: intrinsic quality, contextual quality, presentational quality, and accessibility quality. Taleb et al. [29] develop a framework that validates the critical importance of intrinsic, contextual, presentational, and accessibility dimensions for robust data quality management across various sectors. Intrinsic quality [30] focuses on the essential characteristics of data, which is the basic requirement of data quality, such as integrity, accuracy, consistency. Contextual quality [31] is closely related to data usage scenarios, encompasses considerations like timeliness, task relevance, and applicability. The contextual quality [32] of the same data in different application scenarios may be different. Presentational quality [27] reflects the degree of understandability and conciseness of data. Accessibility quality focuses on the extent to which data can be obtained, and sometimes it also considers factors such as confidentiality and security.

To sum up, intrinsic quality and presentational quality only evaluate the data quality from the aspect of data content. The accessibility quality reflects the difficulty of data acquisition. Only contextual quality remains intricately connected to the specific data application scenario.

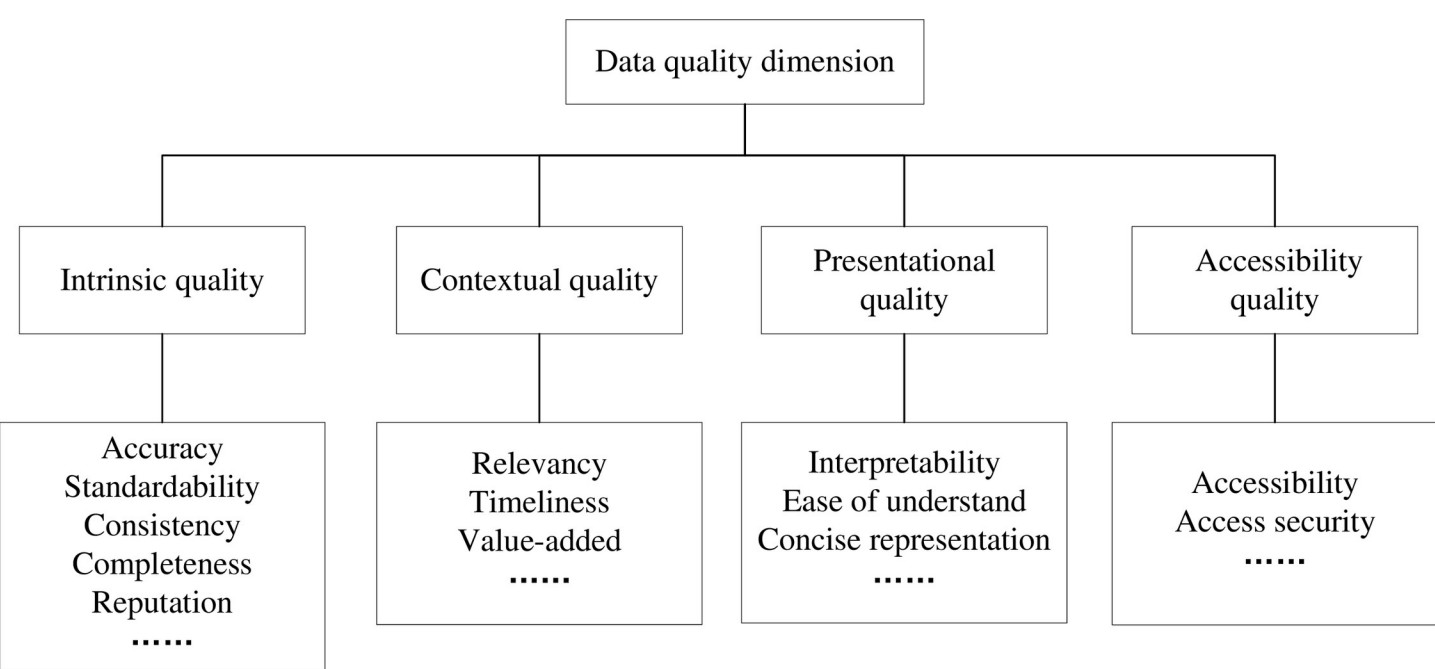

**Fig 1. Data quality dimension classification.**

Nevertheless, the primary emphasis continues to be placed on the fact of data content, particularly with regard to its alignment with the predetermined task, adherence to temporal requirements, and overall accessibility. In essence, the current data quality dimension mainly focuses on data content. However, in the specific application scenario, in addition to the content, the behavior between data and the application scenario (referred to as data behavior) should also be considered. Data behavior can reflect the degree of *participation* and *support* of data to the current scenario or task, as well as its role and value of data.Wang et al. [33] explores how data quality varies according to its use and context, affecting operational efficiency and decision-making. We define this metric as a new contextual quality dimension, data contribution. Data contribution can often provide reference information and a basis for application system resource deployment optimization, business process optimization and reasonable data pricing [34, 35]. Therefore, exploring how to scientifically and reasonably evaluate data contribution holds significant practical importance.

At present, there are few similar related evaluation studies. Our previous work [36] proposes a business scenario-driven data contribution evaluation method based on data access behavior for the single microservice system running independently. The method considers the importance of business scenarios and the corresponding data access behavior information when evaluating the contribution. The importance weight of business scenarios is determined by the analytic hierarchy process, and the information of data access behavior is obtained by the distributed tracing tool. The way to combine the two is to formalize the data access behavior by defining behavior operators so as to express the importance of business scenarios with data contribution, and then use the maximum vector similarity to transform the problem into a nonlinear programming problem. By solving this optimization problem, the contribution of the business-related data tables (databases) in the system to the upper business scene is evaluated.

Data contribution is a contextual quality dimension based on data behavior [37, 38]. At present, data, as a critical production factor and material, has been deeply integrated with all aspects of social life. On the one hand, data behavior provides support for upper business scenarios within the application system [39]. On the other hand, in order to give full play to the value of data and promote industrial collaboration and innovation, data behavior widely exists between different organizations and application systems to support data sharing and exchange [40]. In our first work, we studied the data contribution evaluation method based on data access behavior within a single application system, represented by the microservice system. Therefore, this paper studies the data contribution evaluation method based on data sharing and exchange behavior between different application systems.

The fundamental prerequisite for realizing the full potential of data through sharing and exchange is the assurance of data privacy and security. Departing from the traditional centralized sharing and exchange model, which requires storing shared data in third-party data centers or cloud platforms, the decentralized sharing and exchange model, based on blockchain technology, has gradually become the dominant paradigm. This shift is attributable to its inherent features of openness, tamper-proof design, auditability, and traceability [36]. Consequently, this decentralized approach has found extensive applications in diverse sectors, including government affairs, enterprises, and healthcare [41–44]. According to the blockchain industry survey in 2023 [45], 35% of the domestic organizations have applied blockchain technology in the digital identity field, and 74% of the units are actively exploring the direction of Web 3.0 and digital assets. By the end of 2022, there have been over 1,500 blockchain application cases. The openchain network released by ANT GROUP includes 26 nodes, with more than 1 billion transactions on the chain [46]. Blockchain technology is significantly shaping the future of digital identity and asset management, with a growing number of applications and widespread interest in the potential of Web 3.0. Therefore, this paper chooses the decentralized sharing and exchange mode based on blockchain as the research scenario. The characteristics of this scenario can be summarized as "authorization on the chain, sharing and exchange off the chain" [47, 48]. Fig 2 shows the general process, which mainly includes the processes of sharing data release on the chain, request initiation, authorization verification, data encryption and transmission, acquisition calculation results, and sharing and exchange metadata information to be recorded on the chain. Specifically, the metadata information of the data sharing and exchange behavior between the participants is recorded on the blockchain in the order of timestamp, usually including timestamp, requester, provider and corresponding database, data digest hash, data volume, etc. It is worth noting that in contrast to completely open and untrusted scenarios of digital currencies such as Bitcoin and Ethereum, the identities of participants engaged in data sharing and exchange are mostly known. This implies that it is not a completely untrusted scenario, but rather involves data sharing and exchange within specific contexts, such as various government departments or subsidiaries within a corporate group. Consequently, these scenarios often utilize consortium blockchains [49] to facilitate data sharing and exchange among participants, and uses traditional consensus algorithms such as Raft [50] and PBFT [51] to achieve consensus. Elouataoui et al. [52] discusses the integration of advanced data sharing technologies like blockchain, which enhance data integrity and trust, indirectly supporting the use of these technologies in ensuring data quality and security.

In this paper, we consider two kinds of sharing and exchange behaviors. The first type involves the temporary sharing of data usage rights. For instance, in the open sharing scenario of government data, a provident fund management entity might request access to current tax and credit information of customers. The second type pertains to multi-party computing scenarios. For instance, in a corporate group setting, four subsidiaries may each contribute a

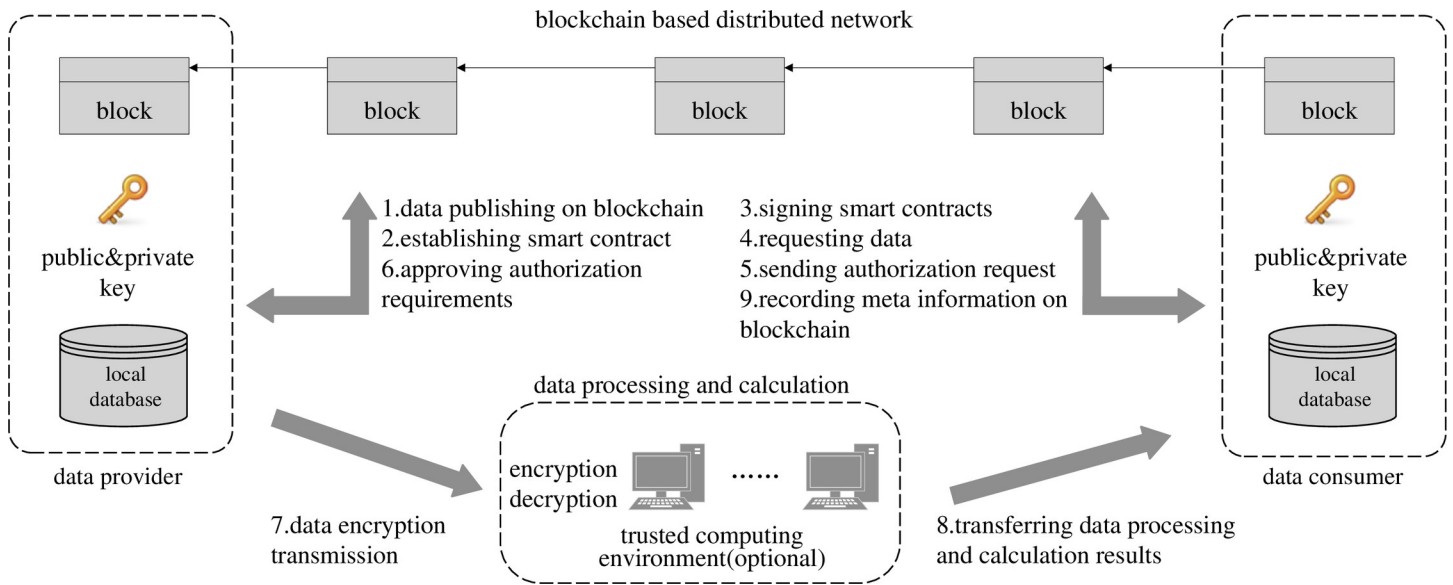

**Fig 2. The process of data sharing and exchange based on blockchain.**

portion of their own data to participate in a computational task. Upon completion of the task, each party receives its respective computation results. Furthermore, since sharing and exchange behaviors involving the same participating database are linked to the same specific databases in the assessment framework, our study defines all sharing and exchange behaviors participated by the same databases as the same type of behavior paradigm. The shared data directory on the blockchain records information such as databases available for sharing and the associated access permissions provided by various organizations. It encompasses all data sharing and exchange behavior paradigms that may occur among participants.

From the perspective of real-world scenarios, the data sharing and exchange behaviors among different application systems mentioned above are often driven by the collaborative business between them. Therefore, there typically exists a certain logical order and dependency relationship between the sharing and exchange behavior paradigms belonging to the same collaborative business. The logical order and dependency relationships reveal the varying degrees of influence and importance among different behavior paradigms. Consequently, they can impact the magnitude of contributions to the associated databases. We aim to obtain this relationship from the metadata of sharing and exchange behavior on the blockchain as a basis for evaluating the contribution of each database. However, in the consortium blockchain system, the data sharing and exchange behavior paradigm is determined and limited, and the same behavior paradigm usually belongs to multiple different collaborative businesses; At the same time, the metadata information of sharing and exchange behavior is recorded chronologically, and the concurrent execution of multiple collaborative services will lead to the continuous sharing and exchange behavior in a short period belonging to different collaborative services; The sharing and exchange behaviors of complex collaborative services usually involves different data-requesting parties. These characteristics make it challenging to mine the dependence between sharing and exchange behavior paradigms. In addition, quantifying the contribution of each participating database according to the dependence of behavior paradigm and data sharing and exchange behavior information is also a problem to be solved in this paper.

To solve the above problems, we propose a data contribution evaluation method based on maximal sequential patterns of behavior paradigms (DecentralDC). The method is divided into three stages. Firstly, the data behavior metadata sequence recorded on the blockchain is preprocessed to obtain the behavior paradigm sequence set, followed by mining maximal sequential patterns of behavior paradigms. Secondly, a behavior paradigm dependency graph is constructed based on the extracted maximal sequential patterns. This dependency graph is a weighted directed graph, and the importance weight of each behavior paradigm can be obtained through the importance evaluation algorithm of graph nodes. Lastly, leveraging the importance weights of behavior paradigms and the data volume of sharing and exchange behaviors, the contribution of databases participating in sharing and exchange over a specific period can be quantified. Our main contributions can be summarized as follows:

- For the decentralized sharing and exchange scenario of multiple application systems based on blockchain, a data contribution evaluation method based on behavior paradigm maximal sequential patterns is proposed.

- The importance weight of the behavior paradigm is determined by the behavior paradigm dependency graph, and the data volume of sharing and exchange behavior is taken into account when quantifying the data contribution.

- Through experiments conducted in two different scale scenarios, this paper verifies the effectiveness of the proposed method and the positive impact of introducing data contribution on data pricing. Specifically, in terms of mining behavior paradigm dependencies, compared to the baseline methods, the F1 scores of our method in the two scenarios are on average higher by 0.15 and 0.11, respectively. Regarding the influence on data pricing, compared to the cumulative regret and regret rate of the baseline method, our method shows a decrease of 43.38% and 61.8% in cumulative regret and regret rate, respectively, in Scenario #1 with the introduction of contribution. In Scenario #2, the corresponding decreases are 35.46% and 59.8%, respectively.

## Related work

While the related works on data contribution is relatively limited, we can draw insights from mature methods in the field of node importance assessment. Node importance evaluation is a critical topic in network analysis, aimed at identifying key nodes within a network. These methods are typically categorized into three types: iterative-based algorithms, path-based algorithms, and adjacency-based algorithms. Each method considers different perspectives and techniques in assessing node importance, thus they each have their own advantages and disadvantages. Table 1 is a comparison table outlining the strengths and weaknesses of these methods.

Relative to traditional methods (iterative algorithms, path-based algorithms, and adjacency-based algorithms), our DecentralDC method offers unique advantages as follows:

- Regarding Iterative Algorithms: Our DecentralDC method resolves issues of high computational complexity and poor convergence by employing maximal sequential pattern mining, eliminating the need for repeated network traversals and significantly boosting efficiency.

- Regarding Path-Based Algorithms: Unlike traditional methods that rely on static paths and struggle with large-scale networks, DecentralDC uses behavioral pattern analysis to uncover complex and dynamic inter-node dependencies, accurately reflecting the real significance of nodes in dynamic network environments.

**Table 1. Comparison of data usage right sharing and multi-party computation.**

| Category | Algorithm | Applications | Key Features | Advantages | Limitations |
|---|---|---|---|---|---|
| Iterative Algorithms | PageRank [53] | Web ranking, search engines | The importance of a page is determined not only by its direct connections but also by the importance of those pages. | • Capturing the global structure and complex interdependencies among nodes.<br>• Adapting to dynamic changes in network structure, updating node importance in real time. | • High computational demand, especially in large networks<br>• May face slow or non-converging issues in frequently changing or complex networks. |
| | Katz Centrality [54] | Social network analysis, dense networks | Considering both direct and indirect connections | • Offering a global perspective on node importance. | • Considering both direct and indirect connections of nodes<br>• providing a global perspective on node importance. |
| Path-Based Algorithms | Betweenness Centrality [55–57] | Traffic networks, information propagation | Evaluating node control by frequency of appearance on shortest paths between all node pairs. | • Precisely identifying and measures a node's control over the network flow.<br>• Especially suitable for assessing key nodes in the flow of information or resources. | • High computational cost, especially when analyzing all paths in large networks.<br>• Overlooking potential network dynamics and non-shortest path influences. |
| | Closeness Centrality [58] | Network efficiency analysis, emergency response systems | Measuring the centrality of a node by its average shortest distance to all other nodes in the network. | • Analyzing node's strategic position and role through path dependency | |
| Adjacency-Based Algorithms | Degree Centrality [59, 60] | Social networks, collaborative networks | Directly calculating the number of connections for each node, reflecting its activity level and influence. | • Simple implementation, easy to understand and compute<br>• Efficient for large-scale data, quickly providing a preliminary assessment of node importance | • Only considering direct connections, overlooking indirect or long-distance impacts<br>• Not fully reflecting a node's influence and authority across the entire network |
| | Eigenvector Centrality [61] | Network influence analysis, recommendation systems | Not only considering direct adjacency but also the quality of those connections. | • Considering the importance of a node's neighbors<br>• Considering node connected to many high-centrality nodes | |

• Regarding Adjacency-Based Algorithms: By constructing behavior paradigm dependency graphs and utilizing maximal sequential pattern mining, our DecentralDC transcends the limitations of focusing solely on local neighbor information, enabling a comprehensive global analysis of node dependencies.

Through these enhancements, DecentralDC markedly improves both the accuracy of node importance assessments and the adaptability of networks, proving particularly effective in scenarios involving data sharing and exchange. In essence, DecentralDC significantly enhances node importance assessments' accuracy and network adaptability. It proves particularly effective in data sharing and exchange scenarios, providing new avenues for network analysis research.

## Preliminaries

This section introduces the prior knowledge needed in this paper, mainly related to the theory of sequential pattern mining and data pricing.

## Sequential pattern mining

Sequential pattern mining [62] is an important research direction in the field of data mining, which aims to mine frequent subsequence patterns in sequence database. Based on the original association rules [63], it considers the time or space dimension information. It has been widely used in user behavior pattern analysis [64], DNA sequence analysis [65], traffic warning [66], etc. The basic concepts involved in sequential pattern mining are as follows.

**Itemset**. A set consisting of one or more items is called an itemset, which can be expressed as $I = \{i_1, i_2, \ldots, i_n\}$. The items in the itemset cannot be repeated, have no order, and can usually be arranged in lexicographic order.

**Sequence & sequence database**. An ordered arrangement consisting of one or more itemsets is called a sequence, which can be expressed as $S = < s_1, s_2, \ldots, s_n >, s_i (1 \leq i \leq n)$ is an itemset; itemsets in a sequence are also called elements or transactions. The same elements can be repeated in a sequence. The sequence length is defined as the number of elements contained in the sequence; the sequence size is defined as the number of items contained in the sequence (cumulative counting of repeated elements and items). For example, the length of the sequence $< \{1, 4\}, \{3\}, \{1, 4\} >$ is three and the size is five. A set consisting of one or more sequences is called a sequence database and is denoted as $SDB = [S_1, S_2, \ldots, S_n]$.

**Subsequence**. The sequence $subseq = < sub_1, sub_2, \ldots, sub_n >$ is a subsequence of another sequence $seq = < s_1, s_2, \ldots, s_m >$ if there exist integers $1 \leq j_1 < j_2 < \ldots < j_n \leq m$ such that $sub_1 \subseteq s_{j_1}, sub_2 \subseteq s_{j_2}, \ldots, sub_n \subseteq s_{j_n}$ (denoted as $subseq \subseteq seq$).

**Continuous subsequence**. Given a sequence $seq = < s_1, s_2, \ldots, s_n >$ and a subsequence $c$, $c$ is a continuous subsequence of $seq$ if any of the following holds: $c$ is obtained by deleting items in $s_1$ or $s_n$; $c$ is obtained by deleting one item of element $s_i (1 \leq i \leq n)$ containing at least two items in $seq$; $c$ is a continuous subsequence of $c'$, and $c'$ is a continuous subsequence of $seq$.

**Support**. The support of a sequence $seq$ in a sequence database $SDB$ is defined as the proportion of sequences containing $seq$ in the whole sequence database and is denoted by $sup_{SDB}(seq)$.

$$sup_{SDB}(seq) = \frac{|\{S \in SDB | seq \subseteq S\}|}{|SDB|}$$

**Frequent sequential pattern**. For a given sequence set $SDB$ and a specified minimum support threshold $min\_sup$, all sequences in the set $\{seq | sup(seq) \geq min\_sup\}$ are called frequent sequence patterns. If the size of a frequent sequential pattern is $k$, this sequence is called $k$-frequent sequential pattern. Any subsequence of a frequent sequential pattern is also a frequent sequential pattern, and any supersequence of an infrequent sequence is not a frequent sequential pattern.

**Maximal sequential pattern**. A frequent sequential pattern is a maximal sequential pattern iff any supersequential pattern of the sequential pattern is not a frequent sequential pattern. Maximal sequential patterns are compact representations of frequent sequential pattern sets whose size is typically several orders of magnitude smaller than the set of frequent sequential patterns [67].

## Data pricing

A data market is a centralized environment and platform where data transactions occur [68, 69]. A well-functioning data market necessitates the rational pricing of data. The stakeholders involved in data transactions typically include providers, consumers, and brokers. Data providers are individuals or organizations that offer data, and the quality and value of the data

they provide determine the trading price and popularity of the data. Data consumers are individuals or entities needing data, selecting suitable data based on their requirements, and paying the corresponding price. Data agents are the third-party intermediaries that provide data services. They provide technical and service support and reasonable data pricing to promote transactions between data providers and consumers. In decentralized scenarios, the intermediary role of data brokers is always eliminated. Data providers and consumers engage in peer-to-peer (P2P) data transactions, and the data pricing is directly completed by the data providers.

**Market value**. Consider a general online data market where various consumers sequentially submit purchase requests over time. These requests can be regarded as a multi-round online transaction sequence [70]. In round $t(t = 1, 2, \ldots)$, the characteristics of the transaction data can be described by a feature vector $\boldsymbol{h_t} \in R^k$, where the actual market value of data $v_t^*$ exists objectively but is unknown. However, the estimated market value $v_t$ can be defined as a function of the feature vector, i.e., $v_t = f(\boldsymbol{h_t}) + \delta_t$, where $f: R^k \rightarrow R$ represents the deterministic part of mapping the feature vector $\boldsymbol{h_t}$ to its value, and $\delta_t$ is a non-negative random variable, assumed to be i.i.d. drawn from a zero-mean distribution over the set of real numbers $R$ [71–73]. For computational simplicity, we can hypothesize $f$ to be a linear function and remove the influence of randomness $\delta_t$. Consequently, the market value can be simplified as $v_t = \boldsymbol{h_t}^T \boldsymbol{\theta}$ ($\boldsymbol{\theta}$ is the weight of the linear model) [74].

**Cost of data**. The cost of data, denoted as $cost_t$, consists of the expenses incurred in processes such as collection, processing, production, and tracking [68]. To simplify the calculation, the data costs can generally be regarded as the privacy compensation generated by the data provider when collecting data, and it can be computed using the feature vector $\boldsymbol{h_t}$ of the data. Moreover, $cost_t$ is typically not higher than the data's posted price $p_t$ to ensure non-negative data utility. Only when the posted price $p_t$ exceeds $cost_t$ can the data provider attain profits [75].

**The target of data pricing**. During online transactions, the regret at round $t$ $R_t$ [76] is defined as the gap between the data provider's maximum expected profits and their actual profits:

$$R_t = \begin{cases} 0, & \text{if } cost_t > v_t \\ \max_{p_t^*}(p_t^* \cdot \Pr(p_t^* \leq v_t) - p_t \cdot \mathbf{1}\{p_t \leq v_t\}) & \text{otherwise} \end{cases} \tag{1}$$

where $\mathbf{1}\{p_t \leq v_t\}$ is an indicator function. When $p_t \leq v_t$, the result is 1; otherwise, it is 0. $p_t^* \cdot \Pr(p_t^* \leq v_t)$ is the data provider's expected profits when the posted price is $p_t^*$, and $p_t^*$ is the optimal posted price that maximizes the profit at round $t$. If the best posted price is $v_t$, (i.e., $p_t \leq p_t^* = v_t$), then regret $R_t$ becomes [77]:

$$R_t = \begin{cases} 0, & \text{if } cost_t > v_t \\ v_t - p_t \cdot \mathbf{1}\{p_t \leq v_t\} & \text{otherwise} \end{cases} \tag{2}$$

Generally, the target of data pricing is to minimize cumulative regret in data transactions:

$$\min \sum_t R_t \tag{3}$$

where $R_t$ is the regret at round $t$.

**Principles of data transactions**. Based on the objectives of data pricing, the following principles can be derived [78–80]:

If $cost_t > v_t$, the transaction will not occur, resulting in zero regret.

If $cost_t < v_t$ and $p_t > v_t$, the regret is $v_t$ (in this scenario, the consumer rejects the posted price, indicating a high likelihood of non-occurrence of the transaction and causing substantial regret).

If $cost_t < v_t$ and $p_t < v_t$, the transaction will occur, resulting in a regret of $v_t - p_t$.

**Posted price**. For the posted price in round $t$, the upper bound $\bar{p}_t$ and the lower bound $\underline{p}_t$ could be set as the upper and lower bound of estimated market value $v_t$, i.e., $\bar{p}_t = \max_{\theta \in \kappa_t} h_t^T \theta$ and $\underline{p}_t = \min_{\theta \in \kappa_t} h_t^T \theta$, where $\kappa_t$ is the knowledge set of $\theta$ in round $t$. There are two methods to choose the appropriate posted price based on $\bar{p}_t$ and $\underline{p}_t$ [77, 81]:

Exploitation price. The exploitation price refers to considering the known optimal price more during each decision-making instance. It selects previously validated high-yield choices. Opting for exploitation pricing as the posted price does not update the knowledge set of $\theta$ (i.e., $\kappa_{t+1} = \kappa_t$). Exploitation pricing primarily emphasizes immediate returns and is typically determined as $max(cost_t, \underline{p}_t)$.

An effective pricing mechanism strives to strike a balance between selecting exploration prices and exploitation prices to achieve maximum revenue. When the difference value between $\bar{p}_t$ and $\underline{p}_t$ exceeds a predefined threshold $\epsilon$, the exploration price is chosen to update the knowledge set of parameters $\theta$. When $\bar{p}_t - \underline{p}_t < \epsilon$, the exploitation price is preferred to ensure successful transactions and attain higher economic gains.

## Problem definition

As shown in Fig 3, given a decentralized data sharing and exchange scenario **FS** = (**P**, **Bx**) based on consortium blockchain, where **P** = ($P1, P2, \ldots, Pr$) represents $r$-number participants

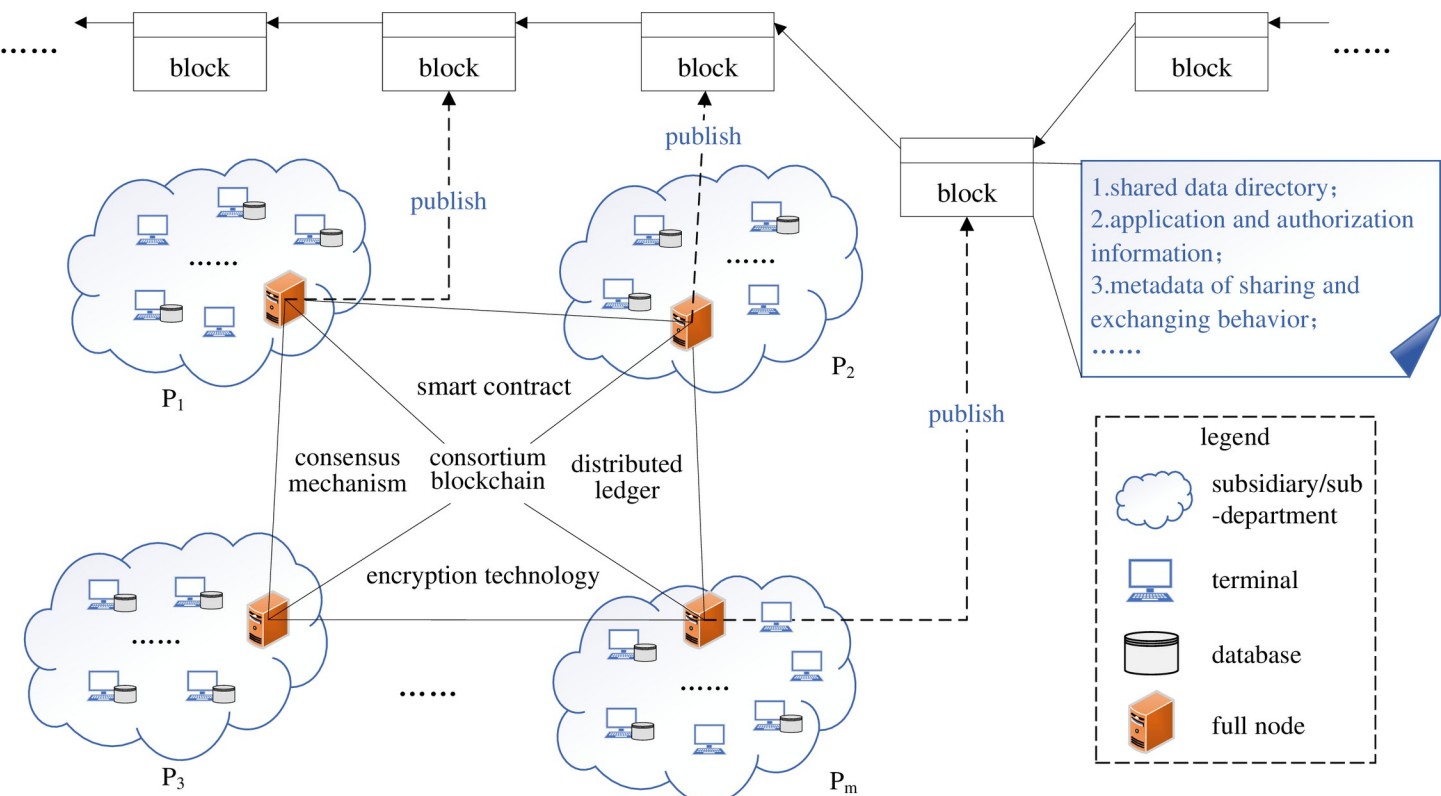

**Fig 3. Scenario diagram for data contribution assessment in decentralized sharing mode.**

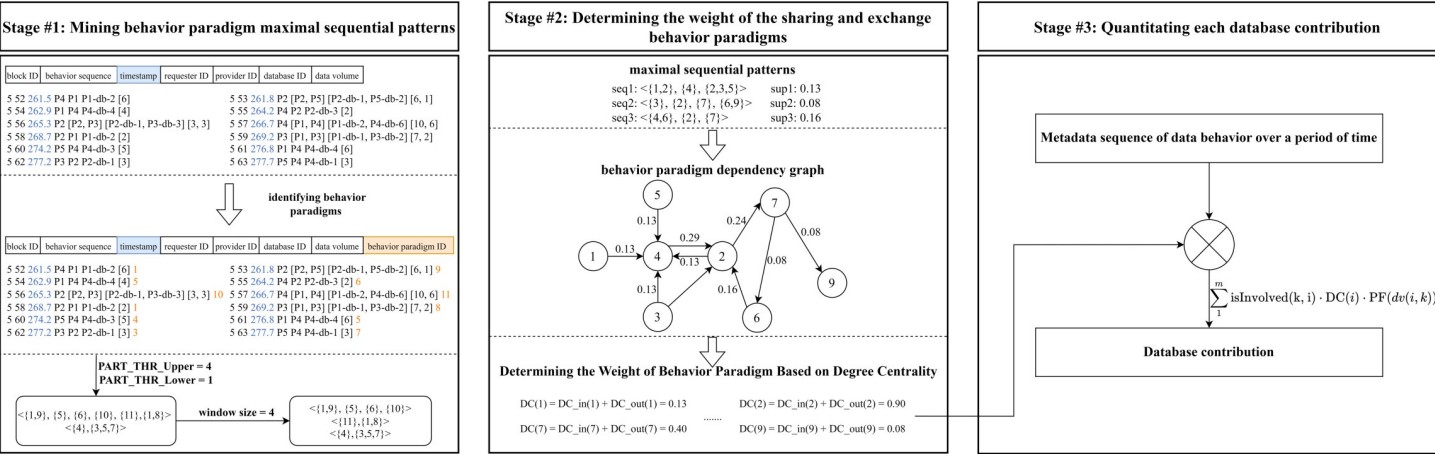

**Fig 4. The architecture of our method.**

in the scenario, while $\mathbf{Bx} = (bx_1, \ldots, bx_m)$ represents the sequence of $m$-number data sharing and exchange behavioral metadata entries recorded on the blockchain over a given timeframe. Each behavioral metadata entry captures details such as the requester, provider, source database, and data volume associated with the respective data sharing and exchange events. We denote the number of databases participating in sharing and exchange of the $i$-th party as $share\_num_i$, i.e., $\sum_{i=1}^{r} share\_num_i = n$, where $n$ is the sum of databases participating in sharing and exchange in the scenario. The task of this paper is to evaluate the contribution vector $\mathbf{y} = (y_1, \ldots, y_n)$ of these $n$-number databases to the data sharing and exchange between different application systems based on the behavior information recorded in $\mathbf{Bx}$.

## Methdology

Based on the above task, we propose a data contribution evaluation method based on maximal sequential patterns of behavior paradigms, and the framework is illustrated in Fig 4. The method is divided into three stages. In the first stage, we preprocess the original sharing and exchange behavior metadata sequence to obtain the behavior paradigm sequence set, and then identify maximal sequential patterns utilizing the maximal sequential pattern mining algorithm. In the second stage, the behavior paradigm dependency graph is constructed based on maximal sequential patterns. Each node represents a behavior paradigm. Two behavior paradigms in a two-continuous subsequence with length two in a maximal sequential pattern are directly connected by an edge in the dependency graph, and the weight of the edge is the sum of the support of all maximal sequential patterns containing this two-continuous subsequence. Then the degree centrality of each node is used as the importance weight of each behavior paradigm. In the third stage, for the metadata sequence of sharing and exchange behavior over a period of time, the contribution of each database participating in sharing and exchange is calculated and normalized by combining the weight of behavior paradigm and the data volume of the sharing and exchange behavior.

As show in Algorithm 1, the main function orchestrates the complete data contribution evaluation process from start to finish, ensuring adherence to the theoretical constructs outlined in the research. The function initializes by obtaining a sequence of behavior metadata FS, which is pre-processed and analyzed through several stages to derive the final contribution scores of databases involved in data sharing and exchange scenarios.

**Algorithm 1** Main Function for Calculating Contribution Scores

```
1: Input: FS—Sequence of sharing and exchange behavior metadata
2: Output: x—Contribution scores for databases
3: // Step 1: Extract maximal sequential patterns from metadata
4: MSP ← MineMaximalPatterns(FS)
5: // Step 2: Construct dependency graph and determine weights
6: W ← ConstructDependencyGraphAndCalculateWeights(MSP)
7: // Step 3: Compute contribution scores using weights and metadata
8: x ← ComputeContributionScores(FS, W)
9: // Output the calculated contribution scores
10: Output x
```

## Mining behavior paradigm maximal sequential patterns

The core of our method involves mining the dependencies between behavior paradigms and determining the weights of these paradigms. Since these dependencies are embedded in the maximal sequential patterns of behavior paradigms, the first step is to mine these maximal sequential patterns.

**Constructing the set of behavior paradigm sequences.** To carry out the subsequent sequential pattern mining task, it is necessary to preprocess the original data sharing and exchange behavior metadata sequences to construct the behavior paradigm sequence set. The overall process is shown in Fig 5. Initially, we utilize the data provider and database provided by it as the identifier to differentiate sharing and exchange behavior paradigms, and assign the same behavior paradigm ID to label each category of behaviors. As illustrated in the example of Fig 5, the behavior paradigm with the ID of 6 represents the category of sharing and exchange behaviors with the provider of "P2" and participating database of "P2-db-3". Then, two interval thresholds (PART_THR_Lower and PART_THR_Upper) are set to segment the behavior pattern ID sequence. The segmentation process involves scanning each behavior chronologically. If the time interval between two adjacent behaviors is less than PART_THR_Lower, their behavior paradigm IDs form an itemset. If the interval exceeds PART_THR_Upper, the behavior paradigm sequence is split between the two behaviors, resulting in two separate sequences. Otherwise, the current paradigm ID is added as an individual itemset to the preceding adjacent sequence. After segmentation, several extended sequences of behavior paradigms are obtained from the original dataset. Finally, behavior paradigm sequences are processed employing a sliding window with a step size equal to the window length, forming the final behavior paradigm sequences set.

**Mining maximal sequential patterns of behavior paradigms.** As mentioned above, frequent sequential pattern mining aims to uncover frequent subsequence patterns within extensive sequences, revealing associative relationships among different itemsets. In this paper, the frequent sequences of behavior paradigms contain the logical order and dependence between behavior paradigms. Note that the dependencies in our study only refer to direct dependencies, excluding transitive and indirect dependencies. However, due to the property that all subsequence patterns of frequent sequential patterns are also frequent, numerous dependencies obtained by typical mining algorithms are redundant or inaccurate. Specifically, continuous subsequences within frequent sequences contain redundant dependencies, while non-continuous subsequences may include erroneous dependencies. Consider the frequent sequence *seq* in Fig 6, which is assumed to depict a collaborative business process. The highlighted red 2-subsequences become inaccurate if not belonging to the direct dependency in any other collaborative business.

As a variant of frequent sequential pattern mining, maximal sequential pattern mining [82–84] can effectively solve the above problems. This approach preserves the longest sequential

A fragment of sharing and exchange behavior metadata sequence

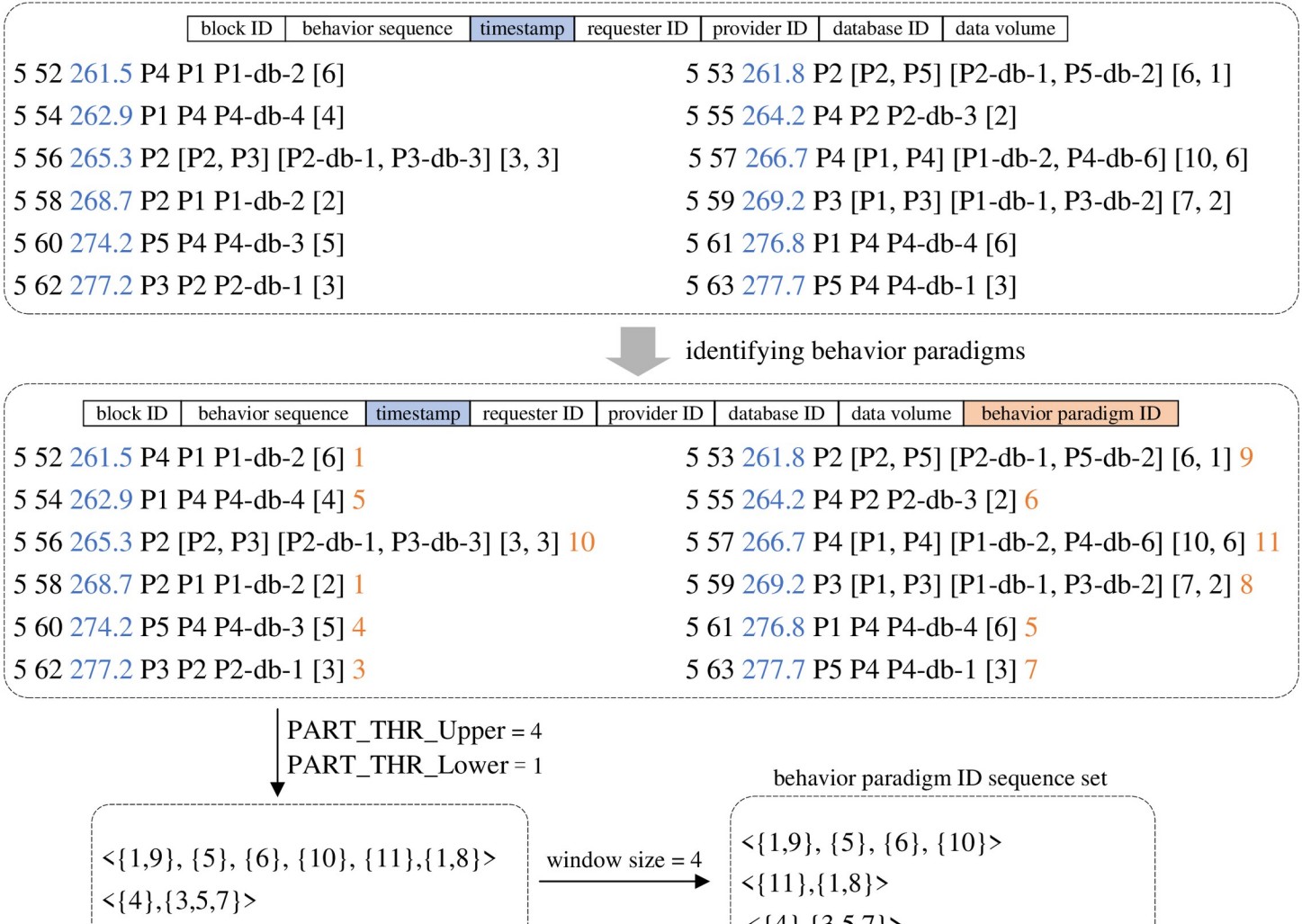

Fig 5. Schematic diagram of behavior paradigm sequence set construction.

pattern meeting support criteria, substantially mitigating the redundancy of frequent sequential patterns and the erroneous dependencies. Moreover, the longer behavior paradigm sequence can contain more complete and rich business logic meaning. In this study, we use the VMSP (Vertical mining of Maximum Sequential Patterns) algorithm [67] of Professor Philippe Fournier-Viger's team to mine maximal sequential patterns of behavior paradigms. This algorithm, identified as the fastest among the their proposed maximal sequential pattern mining algorithms, operates on a vertical mining approach.

This process is integral to identifying and extracting maximal sequential patterns, which are pivotal in understanding the underlying collaborative business in data sharing. These patterns provide the foundation for further analysis in weight calculation. The pseudocode is shown in Algorithm 2.

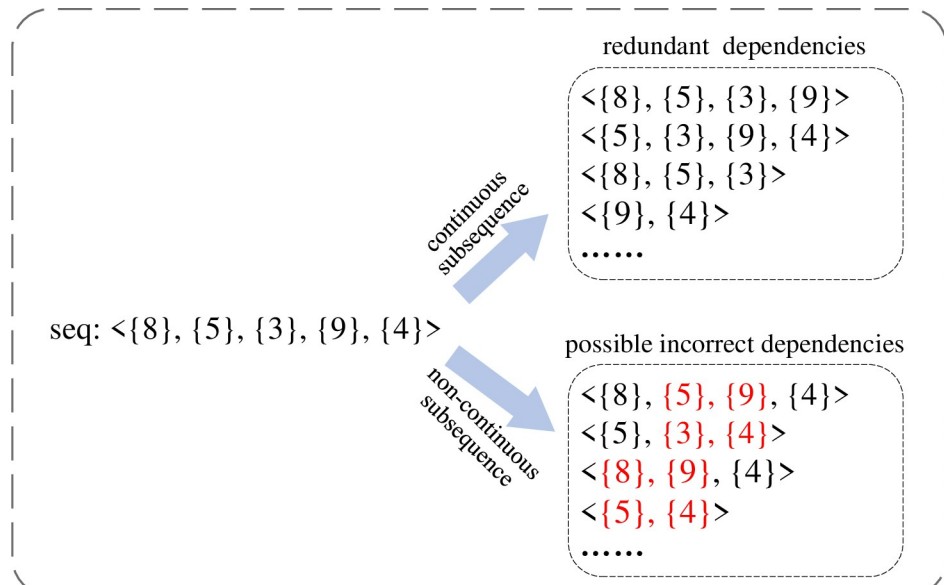

**Fig 6. Two cases of frequent sequence subsequences.**

**Algorithm 2** MineMaximalPatterns Function
H
```
1: function MINEMAXIMALPATTERNS(FS)
2:    Input: FS—Sequence of sharing and exchange metadata
3:    Output: MSP—Maximal Sequential Patterns
4:    P ← ∅            ▷ Initialize an empty set for behavior paradigms
5:    for each bx in FS do
6:      Bx_paradigm_id ← IDENTIFYPARADIGM(bx.provider, bx.database)
7:      P ← P ∪ {Bx_paradigm_id}        ▷ Collect unique paradigms
8:    end for
9:    MSP ← MAXIMALPATTERNMINING(FS, P)        ▷ Apply pattern mining
10:     return MSP
11: end function
```

## Determining the weight of the sharing and exchange behavior paradigms

After obtaining the maximal sequential patterns of behavior paradigm, we aim to quantify the weight of each behavior paradigm through the dependency relationship contained in it, which is divided into two steps: Constructing behavior paradigm dependency graph and evaluating the importance weight of dependency graph nodes.

**Constructing behavior paradigm dependency graph.** For the mined maximal sequential patterns of behavior paradigms, we construct a dependency graph to quantify the dependence strength between behavior paradigms and visualize these dependence relationships. In the dependency graph, each behavior paradigm serves as a node and every two-continuous subsequence of length two $seq=<a, b>$ contained in any maximal sequence pattern forms a directed edge $e=<a, b>$. The weight $w_e$ of edge $e$ is calculated as follows:

$$w_e = \sum_{i=1}^{n} \text{isContained}(e, i) * \sup(i) \tag{4}$$

where $n$ represents the number of mined maximal sequential patterns, $\sup(i)$ denotes the support of the $i$-th maximal sequential pattern $MSP_i$, and the function isContained($e,i$) takes

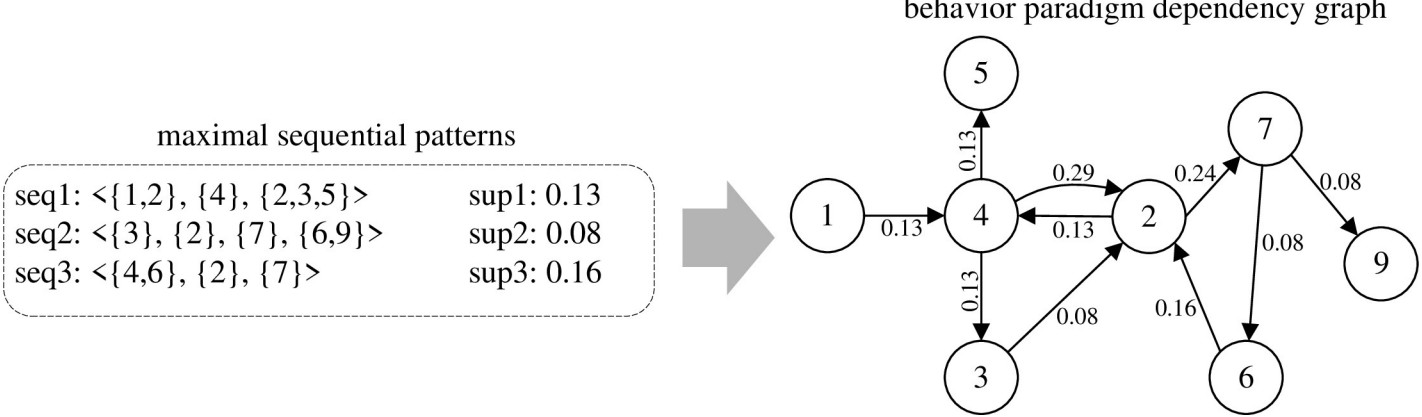

**Fig 7. Schematic diagram of constructing behavior paradigm dependency graph.**

values of 1 or 0, indicating whether the sequence *seq* corresponding to the directed edge *e* is a twp-continuous subsequence of length two of $MSP_i$.

Fig 7 shows an example of constructing behavior paradigm dependency graph. For the edge <4,2>, since <{4},{2}> is the two-continuous subsequence of length two in both *seq*1 and *seq*3, $w_{42}$ is the sum of support of *seq*1 and *seq*3, resulting a total of 0.29. Similarly, the weight $w_{27}$ of edge <2,7> is the sum of *sup*2 and *suq*3, equating to 0.24.

**Determining the weight of behavior paradigm based on degree centrality.** The behavior paradigm dependency graph represents the dependence relationship and degree of dependence between behavior paradigms. Based on this graph, the weight of each behavior paradigm can be calculated by complex network node importance evaluation algorithms.

In contrast to the limitations of general degree centrality [85], which predominantly considers local neighbor information, degree centrality (DC) in our method reflects both the number of sequential patterns in which the current behavior paradigm is located and the information about the number of neighboring nodes. Hence, it serves as a suitable evaluation metric for determining the weight of the behavior paradigm. Specifically, for a node *v* in the dependency graph, its degree centrality DC(*v*) can be defined as:

$$DC(v) = DC\_in(v) + DC\_out(v) \qquad (5)$$

where DC_in(*v*) and DC_out(*v*) are in-degree centrality and out-degree centrality of node *v*, respectively.

$$DC\_in(v) = \sum_i^{neighbor_i} w_{iv} \qquad (6)$$

$$DC\_out(v) = \sum_i^{neighbor_j} w_{vj} \qquad (7)$$

where $neighbor_i$ and $neighbor_j$ refer to all nodes directly arriving at *v* and nodes directly arriving from *v* respectively.

Constructing the dependency graph involves mapping the relationships between different behavior paradigms as extracted from the maximal patterns. The weights calculated from this graph quantify the influence or importance of each paradigm in the context of data sharing. The pseudocode is shown in Algorithm 3.

**Algorithm 3** ConstructDependencyGraphAndCalculateWeights Function

1: **function** CONSTRUCTDEPENDENCYGRAPHANDCALCULATEWEIGHTS (MSP)

```
2:    Input: MSP—Maximal Sequential Patterns
3:    Output: W—Weights of behavior paradigms
4:    G ← (V, E) where V ← ∅, E ← ∅ ▷ Initialize graph with no vertices
      or edges
5:    for each pattern p in MSP do
6:      for each consecutive subsequence (a, b) in p do
7:        if ¬∃ edge (a, b) in E then
8:          E ← E ∪ {(a, b)}          ▷ Add edge if it does not exist
9:        end if
10:       update weight (a, b) based on sum of supports containing (a, b)
11:     end for
12:   end for
13:   W ← CALCULATENODECENTRALITY(G)      ▷ Calculate weights based on
      node centrality
14:   return W
15: end function
```

## Quantifying each database contribution

For the specific sharing and exchange behavior between different participants, the behavior paradigm it belongs to identifies the specific databases providing data and the metadata also records the amount of data supplied by these databases. Our method quantifies the contribution of participating databases by considering both the data volume associated with the behavior and the importance weight of the corresponding behavior paradigm. Specifically, for the given sharing and exchange behavior metadata sequence $\mathbf{Bx} = (bx_1, \ldots, bx_m)$ over a period of time, the contribution $y_k$ of the $k$-th participating database in the scenario is calculated:

$$y_k = \sum_{i=1}^{m} \text{isInvolved}(k, i) \cdot \text{DC}(i) \cdot \text{PF}(\text{dv}(i, k)) \tag{8}$$

where $\text{DC}(i)$ represents the weight of the behavior paradigm corresponding to $bx_i$, and $\text{dv}(i, k)$ denotes the volume of data provided by the $k$-th participating database in behavior $bx_i$. The function $\text{PF}(\cdot)$ discretizes data volume proportionally into integers within the interval $[1, r_{\text{high}}]$:

$$\text{PF}(dv) = \min\left( \left\lfloor \frac{dv}{\text{dv}_{max}/r_{high}} \right\rfloor + 1, r_{high} \right) \tag{9}$$

where $\lfloor \cdot \rfloor$ means round down operator, $r_{\text{high}}$ is a predefined integer greater than 1, and $dv_{\text{max}}$ represents the maximum data volume in the training dataset. The function $\text{isInvolved}(k, i)$ indicates whether the current behavior $bx_i$ involves the $k$-th participating database, with its value being either 0 or 1. Thus, the contribution vector $\mathbf{y} = (y_1, \ldots, y_n)$ of the $n$ databases participating sharing and exchange can be obtained, and we take $\frac{\mathbf{y}}{\|\mathbf{y}\|_1}$ as the final contribution of these databases.

The final step involves calculating the contribution scores for each database based on the weighted paradigms and the volume of data they handled. This score quantifies the impact or value of each database's participation in the Sharing and exchange. The pseudocode is shown in Algorithm 4.

**Algorithm 4** ComputeContributionScores Function

```
1: function COMPUTECONTRIBUTIONSCORES(FS, W)
2:    Input: FS—Sequence of sharing and exchange metadata, W—Weights of
      behavior paradigms
```

```
 3:  Output: x—Contribution scores for databases
 4:  x ← Vector initialized to 0           ▷ Start with zero
     contribution
 5:  for each bx in FS do
 6:    db ← bx.database
 7:    volume ← bx.data_volume
 8:    paradigm ← bx.paradigm_id
 9:    x[db] ← x[db] + W[paradigm] × volume      ▷ Compute weighted
      contribution
10:   end for
11:   Normalize x    ▷ Ensure scores are proportionate and comparable
12:   return x
13: end function
```

## Theoretical analysis

The evaluation of data contribution in decentralized sharing and exchange scenarios necessitates a robust theoretical foundation to accurately assess the influence and importance of different data behaviors. This section delves into the theoretical underpinnings that guide our proposed DecentralDC method.

### Maximal sequential pattern mining and data contribution

Maximal Sequential Pattern Mining (MSPM) is used in data mining to identify subsequences in a dataset that occur frequently and are not contained in any larger pattern. This is critical for evaluating data contribution because it reveals key behavioral patterns in the data that play a significant role in data sharing and exchange.

To achieve maximal sequential pattern mining, we first define the concept of support, which is the frequency of an itemset in the dataset. Given a minimum support threshold *minsup*, a sequence $X$ is considered frequent if its frequency in the dataset is not less than *minsup*.

$$\text{sup}(X) = \frac{|\{T \subseteq D : X \subseteq T\}|}{|D|} \tag{10}$$

where $D$ is the set of all transactions in the dataset, and $T$ is a single transaction. Next, to identify maximal sequential patterns, we further check if these frequent sequences are contained in larger frequent sequences. Only those frequent sequences that are not contained in any larger frequent sequences are considered maximal sequential patterns.

$$\text{MSP}(X) = \text{True if } \forall Y(X \subset Y \Rightarrow \text{sup}(Y) < \text{minsup}) \tag{11}$$

This process ensures that only the most representative and informative patterns are retained, avoiding redundancy. In data contribution evaluation, this method effectively identifies the behavior patterns that contribute most to data sharing and exchange. Through maximal sequential pattern mining, we can extract the most informative patterns from the dataset. These patterns represent the most common and important behaviors in the data. Therefore, identifying these patterns helps us understand the overall dynamics of the data and evaluate the contribution of individual data behaviors. This method ensures the accuracy and efficiency of the analysis, focusing only on the most critical behavior patterns and avoiding the processing of redundant data.

Based on the above theory, we can apply maximal sequential pattern mining to data contribution evaluation. First, preprocess the sharing and exchange behavior data on the blockchain to obtain behavior paradigms and behavior paradigm sets. Then, use the maximal sequential

pattern mining algorithm to extract maximal sequential patterns from the behavior paradigm set. The specific steps include:

(1) Identifying sharing and exchange behavior paradigms based on data providers and databases.

(2) Assigning behavior paradigm IDs based on timestamps, set time interval thresholds, and divide behavior paradigm ID sequences.

(3) Generating the final behavior paradigm subsequence set using a sliding window.

## Construction of probabilistic models and dependency graphs

Probabilistic graphical models (PGMs) use a combination of nodes and directed edges to clearly represent the conditional dependencies between data items. Quantifying these dependencies is crucial for evaluating data contribution. Bayesian networks, as a typical type of probabilistic graphical model, can be represented by the joint probability distribution as follows:

$$P(X_1, X_2, \ldots, X_n) = \prod_{i=1}^{n} P(X_i | \text{Parents}(X_i)) \tag{12}$$

In our method, a Bayesian network is constructed to represent the dependencies between data behaviors and shared behaviors. By calculating conditional probabilities, we can quantify the strength of these dependencies.

$$w(X \rightarrow Y) = P(Y|X) = \frac{\sup(X \cup Y)}{\sup(X)} \tag{13}$$

By constructing a Bayesian network, we can clarify the dependencies between data behaviors and quantify the strength of these dependencies. This enables us to identify and measure the influence of specific behaviors on other behaviors, thereby accurately evaluating the contribution of individual data behaviors. The use of Bayesian networks ensures a precise representation of dependencies and provides a systematic method to handle uncertainties and complex dependencies.

In the contribution evaluation process, frequent behavior patterns are converted into dependency graphs where nodes represent these behavior patterns and edges represent the dependencies. The weights of the edges are determined by the conditional probabilities associated with these dependencies. To construct the dependency graph, each behavior pattern is represented as a node within the graph. Dependency relationships are illustrated as edges between nodes, where any consecutive subsequence of length two within a frequent behavior pattern is depicted as a directed edge. The weight of an edge is calculated based on the support of the maximal sequential patterns that include this consecutive subsequence.

$$w(e) = \sum_{i=1}^{n} \text{isContained}(e, i) \cdot \sup(i) \tag{14}$$

where $n$ is the number of maximal sequential patterns containing the behavior pattern corresponding to edge $e$, $\sup(i)$ is the support of the $i$-th maximal sequential pattern $MSP_i$, and isContained($e, i$) indicates whether the subsequence corresponding to edge $e$ is contained in $MSP_i$.

### Evaluating node influence in the graph

Degree centrality (DC) is a metric used to measure the importance of a node. The degree centrality of a node $v$ can be expressed as the sum of its in-degree and out-degree.

$$DC(v) = \sum_{u \in \text{neighbors}(v)} w(u \to v) + \sum_{u \in \text{neighbors}(v)} w(v \to u) \tag{15}$$

In the evaluation of data contribution, by calculating the degree centrality of each node, we can quantify its importance in data sharing and exchange.

Additionally, path-based centrality analysis calculates the total weight of all paths from one node to all other nodes, assessing each node's importance and influence in the network.

$$I(v) = \sum_{u \in V, u \neq v} \max_{P \in P(u,v)} \left( \prod_{(x,y) \in P} P(x \to y) \right) \tag{16}$$

By using degree centrality and path-based centrality analysis, we can identify the most influential nodes in the data network. These nodes are key in data sharing and exchange, and understanding their importance helps in optimizing data management resources and improving overall network efficiency. Degree centrality and path-based centrality analysis provide a systematic approach to quantifying node importance, ensuring accurate and consistent results.

The first step in contribution evaluation involves calculating the degree centrality of each node in the dependency graph of behavior patterns. This calculation helps to quantify the importance weight of each behavior pattern based on its connections within the graph. The second step is to calculate the path-based centrality by determining the weight sum of all paths from one node to all other nodes. This evaluation assesses the node's propagation capability and influence within the network, providing a measure of its overall impact. Specific implementation steps of degree centrality calculation are as follows:

$$\text{in\_DC}(v) = \sum_{i \in \text{neighbors}(v)} w(i \to v) \tag{17}$$

$$\text{out\_DC}(v) = \sum_{j \in \text{neighbors}(v)} w(v \to j) \tag{18}$$

where in_DC(v) and out_DC(v) denotes the in-degree centrality and out-degree centrality, respectively. Path-based centrality calculation involves determining the sum of the weights of the most probable paths from each node to all other nodes within the graph. This method quantifies the importance of nodes by evaluating the path weights. By calculating these path weights, the importance of each node can be assessed based on the cumulative influence it exerts through its connections to other nodes. This approach provides a comprehensive measure of node centrality within the network.

## Experiments

In this section, we introduce the experimental datasets and parameter settings, the results and analysis of the main experiment and analysis experiments.

### Experimental datasets and parameter settings

This section provides a detailed description of the experimental dataset construction method and the parameter settings.

**Scenario settings.** To visually illustrate the specific application of the proposed protocol, we have detailed two core scenarios: data usage right sharing and multi-party computing. These scenarios are widely applicable across various industries such as healthcare [86], real estate [87], finance [88], transportation [89], education [90], and municipal services [91]. We have provide detailed case studies to further validate the authenticity and broad applicability of these scenarios. In the study of these two scenarios, we aim to demonstrate the most critical and common application patterns in modern data exchange and processing. The first scenario, data usage right sharing, represents the need for data access and permission management between government and enterprises, highlighting the importance of data privacy and security, which is crucial in a regulation-heavy modern society. The second scenario, multi-party computation, showcases how complex data analysis tasks can be performed while maintaining data privacy, which is particularly prevalent in finance, healthcare, and multinational corporations. By testing our methods in these two typical scenarios, we ensure the broad applicability and practical value of our research results in the real world.

- **Scenario #1: Data Usage Right Sharing**

This scenario involves the open sharing of government data. For example, in a government data open-sharing scenario, a public fund management unit may request other government data such as tax and credit information of the current customer. Such sharing is temporary and mainly revolves around the temporary granting of data access and usage rights between government departments. This sharing model often involves handling sensitive data, making privacy and data security the primary concerns. This mode allows data owners to maintain ownership while authorizing other users to use the data for a specified period. It is particularly applicable where time-limited access to sensitive data is necessary, under stringent compliance and security measures, such as temporary access to medical records or personal financial information.

**1. Medical Data Sharing** [86]: A healthcare provider in the United States employs blockchain technology to temporarily share patient health data with authorized research institutions for cardiovascular research. This system processes approximately one million authorized health record accesses annually, ensuring data security and compliance while supporting medical research needs.

**2. Academic Record Management** [91]: A university in the United States uses a blockchain platform to manage and certify academic achievements and qualifications, allowing educational institutions and employers to verify students' academic records while ensuring data integrity and security. Since 2018, this system has verified over 50,000 academic records, demonstrating the effectiveness of this sharing mode in enhancing the security and efficiency of academic data management.

- **Scenario #2: Multi-Party Computation**

This scenario describes collaborative data analysis tasks among multiple subsidiaries within a group, such as jointly developing new drugs or conducting risk assessments. Multi-party computation technology is used to ensure that data is processed without decryption to protect the data privacy of all parties. Each subsidiary provides its data portion, and after completing the computational task, each party retrieves its respective results. This scenario requires efficient data collaboration mechanisms and data integration capabilities. This mode supports multiple participants in performing computational tasks while ensuring data privacy. It is suited for cross-institutional collaboration, such as supply chain management or financial market analysis, allowing stakeholders to make informed decisions without disclosing sensitive data, thereby protecting their data privacy.

**1. Financial Risk Assessment** [88]: An international banking consortium uses multi-party computing technology for transaction anomaly detection to prevent and identify fraudulent activities. Members of the consortium analyze transaction patterns collectively without sharing specific customer data. Approximately 500,000 transactions are analyzed daily, ensuring customer data privacy while enhancing the ability to monitor fraudulent activities.

**2. Smart Traffic Management** [89]: In Singapore, blockchain technology is used to manage and optimize the city's traffic system, ensuring transparent data sharing and secure management. Through real-time data analysis, participants efficiently carry out urban planning and traffic management, processing over two million data transactions annually. This application showcases multi-party computing's role in improving efficiency and response times in smart city and traffic management initiatives.

- **Applications of Data Contribution Assessment and Systems**

Data contribution assessment plays a crucial role in data sharing, directly affecting data pricing and revenue distribution, and motivating higher-quality data sharing.

**1. Educational Data Analysis** [90]: A university employs a blockchain platform to evaluate the contributions of teachers and students to educational resources, which then informs their incentives. This system allows the educational institution to distribute resources and rewards more equitably, encouraging more active participation in educational activities. Each semester, the platform processes over 10,000 records of educational activities. This case reflects how assessing individual data contributions can optimize the allocation and utilization of educational resources.

**2. Waste Management** [92]: In the UK, cities like London are using blockchain technology to enhance waste management processes. Blockchain helps in tracking the lifecycle of waste, ensuring proper disposal, and promoting recycling. Citizens who correctly segregate waste are rewarded with tokens that can be used for public services. This system not only improves waste management efficiency but also encourages public participation in environmental conservation efforts.

**Scenario comparison.** As shown in Table 2, we compare the core differences between data usage right sharing and multi-party computation in terms of data providers and participants, data scale, interaction frequency, and behavior paradigm quantity.

**Experimental parameter settings.** To validate our method in real sharing and exchange scenarios, we construct two simulation scenarios with different scales involving multiple participants on a consortium blockchain. The sharing and exchange metadata sequences recorded in the blockchain, denoted as $\mathbf{Bx} = (bx_1, \ldots, bx_m)$, serve as our experimental datasets. The specific construction steps are as follows.

**Step 1**. Set the number and name of participants, the total number of databases owned by each participant, and the directory of shared resources (including the number, name, metadata and access permissions of databases for each participant).

**Step 2**. Define data sharing and exchange behavior paradigms that can occur between the parties in the consortium blockchain based on the shared resource directory. Two types of behavior paradigms are defined: data usage right sharing and multi-party computing. The data format and examples of behavior paradigms are shown in Table 3. The example of data usage right sharing indicates that P1, P2 and P4 can apply to use the data in the second database P3-db-2 of participant P3, and the example of multi-party computing illustrates that P1-db-1 and P3-db-2 can perform corresponding multi-party computing tasks.

**Table 2. Comparison of data usage right sharing and multi-party computation.**

| Feature/Scenario | Data Usage Right Sharing | Multi-party Computation |
|---|---|---|
| Main Participants | Government departments | Subsidiaries within a group |
| Purpose of Data Processing | Provide temporary data access for administrative processes | Collaborate to solve complex issues such as drug development or risk assessment |
| Privacy and Security Concerns | High, due to involvement of sensitive government information | High, using multi-party computation to ensure data is processed without decryption |
| Technical Requirements | Data access control and permission management technology | Data encryption and multi-party computation technology |
| Data Sharing Model | One-to-many data permission sharing, usually one party provides data for multiple parties to use | Multiple subsidiaries act as both data providers and users |
| Processing Flow | Generally involves simple data access and use, not complex data processing | Involves integrated data processing, requiring coordination of multi-party data processing logic |
| Data Sensitivity and Protection Needs | Extremely high, as data usually involves personal and government-sensitive information | Extremely high, as it involves trade secrets and personal privacy |
| Data Providers and Participants | Typically, one government department provides data, multiple departments or agencies as data users | Multiple subsidiaries act as both data providers and users |
| Data Scale | Usually small, as it involves specific administrative operation data | Can be very large, involving complex calculations and big data analysis |
| Interaction Frequency | May be low, usually on-demand access | High, as computational tasks require frequent data exchange and synchronization |
| Behavior Paradigm Quantity | Fewer, as data sharing situations are relatively fixed | More, as it may involve various computational tasks and data interaction models |

**Step 3**. Define a certain number of multi-party collaborative businesses, and each collaborative business is composed of the behavior paradigms defined in **Step 2** arranged in a particular order. Simultaneously ensuring that there are no containment relationships between behavior paradigm sequences of different business scenarios, and that each behavior paradigm sequence within a business scenario contains no duplicate behavior paradigms.

**Step 4**. Simulate the concurrent execution process of the collaborative businesses defined in **Step 3** through multithreading. Each thread executes a specific collaborative business and records the corresponding sharing and exchange behavior metadata onto the blockchain. To emulate varying levels of concurrency in real-world scenarios, the execution flow of threads is designed as follows: Starting from time $t_s$, decide whether to execute the collaborative business with a certain probability $p_s$. If it fails to execute, the thread decide again after a short interval. Once the collaborative business initiated, all sharing and exchange behaviors belonging to this business are sequentially executed with a certain probability $p_e$. After completing this collaborative business, the thread will proceed to repeat the above process after a relative long interval. Simultaneously, the metadata information of each

**Table 3. Data format and examples of sharing and exchange behavior paradigms.**

| Data item | Identification | Description |
|---|---|---|
| Behavior paradigm identifier | *bx_paradigm_id* | Unique identification of the behavior paradigms |
| Behavior paradigm types | *bx_paradigm_type_id* | 0: data usage right sharing; 1: multi-party computing |
| Data provider | *provider_id* | – |
| Participant database | *share_db_id* | – |
| Licensor | *permission_id_list* | Participants eligible to participate in the current behavior |
| Sample | Data usage right sharing: [5, 0, P3, P3-db-2, (P1, P2, P4)] | |
| | Multi-party computing: [12, 1, (P1, P3), (P1-db-1, P3-db-2), (P1, P3)] | |

**Table 4. Metadata format and examples of sharing and exchange behavior.**

| Data Items | Block | Sequence | Timestamp | Requester | Provider | Database | Data Volumn | Data Hash |
|---|---|---|---|---|---|---|---|---|
| *Identification* | *block_id* | *bx_seq* | *timestamp* | *requester_id* | *provider_id* | *db_id* | *data_volume* | *data_hash* |
| *bx_i* | 10 | 151 | 543 | P1 | (P3, P4) | (P3-db-1, P4-db-2) | (1,5) | (hash1, hash2) |
| | 472 | 7005 | 23629 | P4 | P2 | P2-db-3 | 9 | hash1 |

completed sharing and exchange behavior is recorded on the blockchain. Finally, when all threads finish, the recorded metadata information of all behaviors is sorted chronologically, resulting in the dataset **Bx** = ($bx_1$, ..., $bx_m$). The format and examples of $bx_i$ are illustrated in Table 4. The behavior sequence number serves as the unique identifier for the behavior, while the data volume field represents discretized results ranging from integers 1 to 10. The data volume field contains multiple elements for behavior with sequence number 151 in Table 4, indicating a multi-party computing behavior.

To verify the efficacy of our method in the above two scenarios, our experiments include both a small-scale and a large-scale scenarios. Specific parameter settings are detailed in Table 5. The function randint (a, b) in Table 5 represents the random generation of integers within a closed interval [a, b]. For **Scenario #1** (data usage right sharing), considering that the data volume involved is usually small but involves sensitive information, a high level of privacy protection and a low interaction frequency are set. This setting helps simulate the characteristics of temporary data access and permission management among government departments. For **Scenario #2** (multi-party computation), due to the involvement of complex data analysis and large data volumes, a larger dataset size and more frequent data exchanges are set. This setting can better simulate multi-party computation scenarios common in industries such as finance and healthcare, which require efficient data processing and strong data privacy protection. For different experimental scenarios, we set corresponding hyper-parameters to ensure that the experiments can accurately simulate real-world data sharing and exchange behaviors.

To further illustrate the dataset features, the specific descriptions and explanations of the above data items are shown in Table 6.

**Experimental data description.** Our experimental dataset covers two main scenarios: data usage right sharing and multi-party computation. The dataset scale and complexity vary based on the characteristics of these scenarios. In Scenario #1, there are fewer participants and simpler data exchanges, resulting in a smaller dataset size, reflecting the characteristics of actual government data processing. In Scenario #2, the complexity of collaborative tasks among multiple subsidiaries results in a significantly larger and more complex dataset to meet

**Table 5. Parameter settings for two experimental scenarios.**

| Item | Identifier | Scenario #1 | Scenario #2 |
|---|---|---|---|
| Number of behavior paradigms per collaborative business | *bx_num_per_bs* | randint(2,5) | randint(2,5) |
| Probability of business execution | $p_s$ | 0.3 | 0.3 |
| Probability of subsequent business behavior execution | $p_e$ | 0.5 | 0.5 |
| Interval after business completion | *exec_wait_interval* | randint(30,40) | randint(60,80) |
| Interval on business non-execution | *bs_interval* | randint(10,20) | randint(20,40) |
| Upper interval threshold | *PART_THR_Upper* | 5 | 5 |
| Lower interval threshold | *PART_THR_Lower* | 1 | 1 |
| Sliding window size | *window_size* | 8 | 8 |

**Table 6. Feature descriptions for experimental scenarios.**

| Item | Feature Description |
|---|---|
| Identifier | Providing a unique identifier for each hyperparameter to ensure data tracking and differentiation. |
| Scenario #1 | Data usage right sharing. |
| Scenario #2 | Multi-party computation. |
| Number of behavior paradigms per collaborative business | Specifying the number of behavior paradigms involved in each collaborative business, set using a random function to demonstrate variability and uncertainty in the experiment. |
| Probability of business execution | Determining the likelihood of initiating collaborative business at a specific time point, which is crucial for the dynamism and responsiveness of the experimental process. |
| Probability of subsequent business behavior execution | Determining the likelihood of continuing a business behavior once it has started, affecting the continuity and persistence of the experiment. |
| Interval after business completion | Defining the waiting time required by the system after completing a business task, which can influence the pace and overall duration of the experiment. |
| Interval on business non-execution | The waiting time for the system if a business is not initiated at a specific time point, affecting the experiment's waiting strategy and time management. |
| Upper interval threshold | Setting these two thresholds for time intervals considered when processing data, which is crucial for data segmentation and time series analysis. |
| Lower interval threshold | Setting these two thresholds for time intervals considered when processing data, which is crucial for data segmentation and time series analysis. |
| Sliding window size | Setting the size of the sliding window used in sequential pattern mining, which is key to controlling the granularity and coverage of data analysis. |

the highly dynamic and intricate data processing needs. Details of data description are illustrated in Table 7.

In the experiment, to ensure comprehensive learning of the dependency and importance weight of the sharing and exchange behavior paradigms, and to accurately evaluate data contribution, we divided the dataset Bx into a training set and a test set at a ratio of 4:1. Specifically, 80% of the data is used in the training phase. Through this training data, we can establish and optimize the dependency relationship model of the sharing and exchange behavior paradigms and calculate the importance weight of each paradigm. The remaining 20% of the data is reserved for the testing phase. Using this test data, we can validate the effectiveness of the model and ultimately output the data contribution evaluation results. This approach ensures consistency and reliability in model training and evaluation across different datasets. Details are illustrated in Table 8.

## Main results

To further verify our method in this experiment, we will assess the contribution of each database in different scenarios, add data contribution assessment experiments, and compare the differences between our DecentralDC method and other baseline methods. By setting predetermined contribution rankings for different databases and controlling their occurrence probabilities in the experimental data generation, we update the two datasets that embodies certain data sharing and exchange behavior paradigms. Initially, we establish two different data sharing and exchange scenarios in a simulation environment and assign predetermined contribution rankings to the databases in each scenario. Databases with higher contributions will have a higher probability of appearing during experimental data generation, while those with lower

**Table 7. Datasets description of two scenarios.**

| Item | Identifier | Scenario #1 | Scenario #2 | Description |
|---|---|---|---|---|
| Number of participants | $r$ | 4 | 50 | Showing the number of entities or organizations participating in each scenario, reflecting the scale and complexity of the experiment. |
| Number of databases involved in sharing and exchange | $n$ | 9 | 189 | Counting the total number of databases involved in data sharing and exchange, which is important for assessing the dataset's breadth and the data foundation of the experiment. |
| Number of behavior paradigms | $bx\_paradigm\_num$ | 14 | 265 | Listing the number of defined sharing and exchange behavior paradigms in each scenario, which affects the diversity of behavior paradigms and the experiment's coverage. |
| Number of collaborative businesses | $bs\_num$ | 10 | 130 | The number of collaborative businesses involved in each scenario, reflecting the business complexity and interaction frequency of the experiment. |
| Size of dataset $Bx$ | $m$ | 5710 | 32859 | The number of metadata sequences of data sharing and exchange behaviors recorded on the blockchain over a period of time. |
| Behavior paradigm sequence set size | $bx\_paradigm\_seq\_set\_size$ | 1051 | 2038 | The size of the behavior paradigm sequence set, which is the basis for evaluating the performance and processing capability of the algorithm. |
| Maximum data volume level | $r_{high}$ | 10 | 10 | The maximum data volume level, used for quantification and standardization during analysis. |

contributions will have a lower probability. Then, we will generate experimental data based on these preset rankings and occurrence probabilities, which will include the roles and frequencies of various databases in specific sharing and exchange behaviors. By analyzing the experimental results, we can validate the consistency between the preset database contribution rankings and the actual observed data usage. This method not only helps us understand the behavioral impact of different databases in a decentralized sharing and exchange model but also tests the effectiveness and accuracy of our data contribution assessment method in practical applications.

**Experimental purpose and scenario definition.** This experiment aims to build a dataset with a certain data sharing behavior pattern by presetting different database contribution rankings and controlling their occurrence probabilities during data generation. The core objective is to validate the rationality of the database contribution rankings and the accuracy of the assessment method.

We define two different data sharing and exchange scenarios. Each scenario simulates a specific type of data exchange pattern, scenario #1 simulate data sharing between government departments, while scenario #2 simulate data exchanges between medical institutions. Based on prior data usage and contribution analysis, assign a predetermined contribution ranking to the participating databases. Databases with higher contributions will have a higher occurrence probability during data generation, indicating more frequent participation in data exchange activities. A probabilistic model is designed to adjust the occurrence frequencies of databases within the generated dataset based on their predefined contribution levels. This method ensures that databases with higher contributions are more active in the simulation, thus mimicking real-world behavioral patterns. Following the settings described above, a simulation

**Table 8. Division of training and test datasets.**

| Item | Identifier | Scenario #1 | | | Scenario #2 | | |
|---|---|---|---|---|---|---|---|
| | | Total | Train | Test | Total | Train | Test |
| Size of dataset **Bx** | $bs\_num$ | 5710 | 4568 | 1142 | 32859 | 26287 | 6572 |
| Behavior paradigm sequence set size | $bx\_paradigm\_ seq\_set\_size$ | 1051 | 841 | 210 | 2038 | 1630 | 408 |

program is employed to generate data encompassing the exchange activities of all participating databases. This data includes timestamps, the databases involved, data items, and their access frequencies. Each scenario is independently generated to maintain consistency and isolation of the scenario settings.

**Baselines and metrics.** In this experiment, we compare several algorithms to validate the performance differences between our DecentralDC method and existing baseline methods. The following algorithms are compared:

- Iterative Algorithms: including PageRank and Katz Centrality, these algorithms iteratively compute the importance of nodes, suitable for assessing node centrality in networks.

- Path-based Algorithms: including Betweenness Centrality and Closeness Centrality, these methods evaluate the influence of nodes based on their positions within paths.

- Adjacency-based Algorithms: including Degree Centrality and Eigenvector Centrality, which primarily assess importance based on the direct connections of nodes. This evaluation does not involve pre-processing through maximal behavior pattern mining.

For evaluation metrics, we utilize MAP, Precision@10, Recall@10, F1@10, and ACC@10 to comprehensively assess the effectiveness of the different algorithms.

$$MAP = \frac{1}{|Q|} \sum_{q=1}^{|Q|} \frac{1}{m_q} \sum_{k=1}^{m_q} Precision(R_{q,k}) \tag{19}$$

Here, $|Q| = 1$, so the formula calculates the average precision for a single query. $m_q$ represents the number of relevant items for that query, and $Precision(R_{q,k})$ is the precision at each cut-off $k$, up to $m_q$.

$$Precision@10 = \frac{\text{Number of relevant items retrieved in top 10}}{10} \tag{20}$$

This measures the proportion of relevant items among the top 10 items retrieved, reflecting the accuracy of the retrieval process for the top results.

$$Recall@10 = \frac{\text{Number of relevant items retrieved in top 10}}{\text{Total number of relevant items}} \tag{21}$$

This evaluates the proportion of the total relevant items that have been retrieved within the top 10 results, providing insight into the coverage of the retrieval system.

$$F1@10 = 2 \times \frac{Precision@10 \times Recall@10}{Precision@10 + Recall@10} \tag{22}$$

The F1 score at 10 combines precision and recall into a single metric by calculating their harmonic mean, providing a balance between precision and recall for the top 10 results.

$$ACC@10 = \frac{\text{Number of correct predictions in top 10}}{10} \tag{23}$$

This metric measures the ratio of correctly predicted relevant items in the top 10 results, providing a straightforward indicator of accuracy at this cut-off.

**Main experimental results and analysis.** The main results of Scenario # 1 is shown in Table 9. In the scenario of data usage rights sharing (Scenario # 1), DecentralDC demonstrates significant advantages. It achieves a MAP of 0.85, indicating its effectiveness in accurately identifying key nodes through an integrated analysis of the nodes' network positions and data

**Table 9. Main Results of Scenario # 1.**

| Methods | MAP | Precision@10 | Recall@10 | F1@10 | ACC@10 |
|---|---|---|---|---|---|
| DecentralDC | 0.85 | 0.84 | 0.83 | 0.83 | 0.85 |
| PageRank | 0.72 | 0.70 | 0.71 | 0.70 | 0.72 |
| Katz Centrality | 0.75 | 0.73 | 0.74 | 0.73 | 0.75 |
| Betweenness Centrality | 0.68 | 0.66 | 0.67 | 0.66 | 0.68 |
| Closeness Centrality | 0.70 | 0.68 | 0.69 | 0.68 | 0.70 |
| Degree Centrality | 0.77 | 0.75 | 0.76 | 0.75 | 0.77 |
| Eigenvector Centrality | 0.79 | 0.78 | 0.77 | 0.77 | 0.79 |

interaction behaviors. The Precision@10 is 0.84, suggesting that DecentralDC can provide the most relevant entries among the top 10 results, which is crucial for quickly accessing key information in a data-sharing environment. The Recall@10 is 0.83, confirming its effectiveness in covering all relevant data nodes and ensuring that no key data is missed. The F1@10 score is also 0.83, showing a good balance between precision and recall, allowing for accurate node identification while minimizing false positives. The ACC@10 is 0.85, indicating that almost all of the top 10 retrieved results are correct, reflecting the high efficiency and reliability of DecentralDC in practical applications. **Comparative analysis with baseline methods in Scenario # 1 are shown as follows**.

- **PageRank**: In the data usage rights sharing scenario, PageRank scored a MAP of 0.72 and Precision@10 of 0.70. PageRank primarily assesses node importance based on the number of incoming links, which may not adequately reflect the true influence of nodes in dynamically changing data usage scenarios. In contrast, DecentralDC not only considers the number of connections but also analyzes their actual roles and efficiency in data usage, thus providing a more accurate assessment of node influence.

- **Katz Centrality**: For Katz Centrality, the MAP was 0.75 and Precision@10 was 0.73. This method emphasizes the influence of indirect connections between all nodes but may not fully capture the rapidly changing patterns of data flow. DecentralDC surpasses Katz Centrality's limitations by analyzing the directions and frequency of data flows, offering a more dynamic and precise analysis of node importance.

- **Betweenness Centrality**: Betweenness Centrality achieved a MAP of 0.68 and Precision@10 of 0.66. It identifies nodes at central positions within the network, which control information flows. However, in data sharing, nodes serving merely as conduits in information flow may not engage directly in data usage. DecentralDC identifies not only key nodes in the information flow but also analyzes their roles in data processing and decision-making, thus providing deeper insights into data usage rights sharing.

- **Closeness Centrality**: Closeness Centrality recorded a MAP of 0.70 and Precision@10 of 0.68. While this metric emphasizes the proximity of nodes to all others within the network, suggesting their efficiency in information transmission, it may not reflect the nodes' actual control over data usage. DecentralDC provides a more comprehensive evaluation by analyzing not only the structural position of nodes within the network but also their roles in data generation, processing, and usage, thereby offering a more precise assessment of influence.

- **Degree Centrality**: In this scenario, Degree Centrality postes a MAP of 0.77 and Precision@10 of 0.75. This method measures the importance of nodes based on their direct connections, reflecting their activity within the network. Although Degree Centrality can

effectively identify active interacting nodes, it overlooks the specifics and quality of these interactions. DecentralDC, by delving deeper into nodes' behavior patterns and analyzing how they control and use data, provides a more comprehensive understanding of node influence.

- **Eigenvector Centrality**: Eigenvector Centrality performes well in the data usage rights sharing scenario with a MAP of 0.79 and Precision@10 of 0.78. This method suggests that a node's importance is influenced not only by its direct connections but also by the importance of its connected nodes. While Eigenvector Centrality effectively captures the cumulative effect of influence within the network, it primarily relies on static network structures. DecentralDC, by analyzing specific behaviors of nodes in data usage—such as requests, processing, and distribution—reveals the real-time and dynamic importance of nodes.

The main results of Scenario # 2 is shown in Table 10. In multi-party computation environments (Scenario # 2), DecentralDC exhibits significant performance advantages. It achieves a MAP value of 0.80, demonstrating excellent identification of key participating nodes. The Precision@10 is 0.79, indicating its ability to accurately recognize nodes that contribute most significantly to computations. The Recall@10 is 0.78, proving its effectiveness in covering all essential data nodes, crucial for ensuring the completeness and accuracy of computation results. The F1 score at 10 is also 0.78, showing a good balance between precision and recall, which ensures that while identifying the correct nodes, the results' relevance and completeness are maximized. The ACC@10 is 0.80, confirming DecentralDC's efficiency and reliability in practical applications, especially in complex scenarios involving multi-party data integration and processing. **Comparative analysis with baseline methods in Scenario # 2 are shown as follows**.

- **PageRank**: Although PageRank performs well in data usage rights sharing, its effectiveness significantly drops in multi-party computation scenarios (MAP 0.65). This is because PageRank mainly assesses static importance based on links, while multi-party computation requires identifying nodes dynamically contributing to the computation process. DecentralDC provides a more comprehensive importance assessment by evaluating not just static links but also the dynamic interactions and data contributions between nodes.

- **Katz Centrality**: While Katz Centrality performs relatively well in data usage rights sharing with a MAP of 0.68, it faces challenges in multi-party computations. It does not adequately handle the dynamic data flow and contributions between nodes. DecentralDC is more effective in capturing real-time data interactions and processing capabilities, allowing for more adaptable evaluations of node importance in multi-party computations.

**Table 10. Main results of Scenario # 2.**

| Methods | MAP | Precision@10 | Recall@10 | F1@10 | ACC@10 |
|---|---|---|---|---|---|
| DecentralDC | 0.8 | 0.79 | 0.78 | 0.78 | 0.8 |
| PageRank | 0.65 | 0.63 | 0.64 | 0.63 | 0.65 |
| Katz Centrality | 0.68 | 0.67 | 0.66 | 0.66 | 0.68 |
| Betweenness Centrality | 0.62 | 0.60 | 0.61 | 0.60 | 0.62 |
| Closeness Centrality | 0.64 | 0.62 | 0.63 | 0.62 | 0.64 |
| Degree Centrality | 0.70 | 0.68 | 0.69 | 0.68 | 0.70 |
| Eigenvector Centrality | 0.72 | 0.70 | 0.71 | 0.70 | 0.72 |

- **Betweenness Centrality**: Despite its advantages in identifying key nodes in data flows, Betweenness Centrality's importance evaluation capability decreases in multi-party computations (MAP 0.62), as merely being a conduit in data flow does not equate to contributing to computation outcomes. DecentralDC surpasses Betweenness Centrality by analyzing how nodes influence final computation outputs, which is particularly crucial in multi-party computations.

- **Closeness Centrality**: Exhibiting a MAP of 0.64, Closeness Centrality shows limitations in multi-party computations. This method, which evaluates the average distance of a node to all others in the network, assumes that closer nodes have a greater impact on data. However, multi-party computations require more than quick data access; they need nodes that significantly contribute to data processing. DecentralDC provides a more accurate evaluation of node importance by analyzing nodes' actual data processing capabilities and contributions, making it particularly suitable for multi-party computations.

- **Degree Centrality**: With a MAP of 0.70 in multi-party computations, Degree Centrality's primary limitation is its focus solely on direct connections of nodes. In multi-party computations, a node's influence is determined not just by its connections but by how it processes and contributes data. DecentralDC offers deeper insights by analyzing the behavioral patterns of nodes in data processing and exchange, enabling the identification of nodes that are crucial in data processing but may not be prominent in traditional measures.

- **Eigenvector Centrality**: Eigenvector Centrality achieves a MAP of 0.72 in multi-party computations, closely matching DecentralDC. This method considers the importance of a node based on the importance of its connected nodes, attempting to capture the cumulative effect of influence within the network. While theoretically effective in reflecting the distribution of influence in a network, Eigenvector Centrality lacks detailed analysis of how nodes actively participate in data processing and multi-party computations. DecentralDC provides a more comprehensive evaluation of influence by considering nodes' data interactions and processing capabilities, which are essential for optimizing the entire computation process in multi-party environments.

Here are the results of data contribution obtained through our DecentralDC, as shown in Table 11. For the 189 databases in Scenario #2, Table 11 shows the top ten databases in contribution. Given the theory of our method, in the process of sharing and exchange, the database

**Table 11. Database contribution of sharing and exchange databases in two scenarios.**

| Serial Number | Scenario #1 | | Scenario #2 | |
|---|---|---|---|---|
| | Database | Database(%) | Database | Contribution (%) |
| 1 | P1-db-5 | 18.7 | P1-db-3 | 3.277 |
| 2 | P4-db-1 | 18.38 | P16-db-2 | 2.538 |
| 3 | P2-db-1 | 15.99 | P13-db-3 | 2.478 |
| 4 | P3-db-2 | 13.56 | P50-db-4 | 2.105 |
| 5 | P4-db-8 | 8.46 | P3-db-2 | 2.033 |
| 6 | P3-db-3 | 6.76 | P4-db-1 | 1.535 |
| 7 | P1-db-3 | 6.57 | P37-db-5 | 1.513 |
| 8 | P3-db-7 | 5.92 | P14-db-2 | 1.373 |
| 9 | P4-db-2 | 5.64 | P18-db-2 | 1.327 |
| 10 | – | – | P33-db-7 | 1.317 |

**Table 12. Effects comparison of mining behavior paradigm dependency.**

| Scenario | Behavior Paradigm Dependency Mining Algorithm | Precision (%) | Recall (%) | F1 (%) |
|---|---|---|---|---|
| Scenario #1 | CM-SPADE | 52.38 | 100 | 68.75 |
| | t-FCSM | 100 | 90.69 | <u>95.12</u> |
| | Our Method | 94.37 | 100 | **97.10** |
| Scenario #2 | CM-SPADE | 65.46 | 100 | 79.13 |
| | t-FCSM | 100 | 88.75 | <u>94.04</u> |
| | Our Method | 95.57 | 100 | **97.73** |

with high importance weight of the behavior paradigm and a large amount of data will have a higher contribution.

## Maximum sequential patterns mining experiments

This section verifies the effectiveness of the proposed method for maximum sequential patterns mining. For the behavior paradigm dependence mining step, the effect comparison between the maximal sequential pattern mining method and the other two methods is shown in Table 12, and the support of the three methods in the two scenarios is set to 11% and 1.7%, respectively. In Scenario #2, the lower support set compared to Scenario #1 is mainly due to the increased parallel execution of collaborative businesses. Consequently, more behaviors are generated within the same duration, allowing for the acquisition of a greater number of behavior paradigm sequences through interval thresholds and sliding windows, ultimately leading to lower support.

**Baseline methods.** To verify the effect of our method in mining dependencies between sharing and exchange behavior paradigms, two baseline methods are selected for comparison.

- **CM-SPADE** [93]: All frequent sequential patterns of behavior paradigm sequence set are mined. CM-SPADE is a highly efficient sequential pattern mining algorithm proposed by Professor Philippe Fournier-Viger's team in 2014.

- **Two-Frequent Continuous Subsequence Mining (t-FCSM)**: This method is designed to mine Two-frequent continuous subsequence patterns with length 2 from the behavior paradigm sequence set.

We refer to the evaluation indicators of machine learning classification tasks, as detailed in Table 13. Note that in the evaluation process, only the presence or absence of dependencies is considered, while the strength of dependencies, namely the weight of edges in the dependency graph is not taken into account.

For each scenario, the F1 value with the best mining effect is marked in bold, and the next best is marked by underlining. According to the experimental results, we can draw the following conclusions.

**Table 13. Evaluation indicators in the experiments.**

| Metrics | Meaning |
|---|---|
| Precision | Number of all correct dependencies mined / Number of all dependencies mined |
| Recall | Number of all correct dependencies mined / Number of all correct dependencies |
| F1 | $(2 \times \text{Precision} \times \text{Recall}) / (\text{Precision} + \text{Recall})$ |

First, in both scenarios, the recall of CM-SPADE method and VMSP method is 1, indicating that both methods have mined all the correct dependencies. However, the CM-SPADE mining algorithm exhibits lower precision. This can be attributed to the retention of a significant number of subsequences, wherein the considerable dependencies, i.e., Two-continuous frequent subsequences with length 2, are erroneous. Second, in both scenarios, the t-FCSM method achieves a precision of 1, with recalls of 90.69% and 88.75%. This suggests all accurate dependencies mined by t-FCSM, but with some omissions. These omissions stems from exclusive focus on mining sequential patterns within adjacent elements of the behavior paradigm sequence. In our experimental setting, where the probability of the next behavior belonging to the same collaborative business is 0.5, dependencies may be overlooked if two adjacent behavior paradigms in the collaborative business process do not correspond to adjacent elements in the sequence. Moreover, t-FCSM can only mine short binary dependencies, constraining its expression of the business logic meaning. Therefore, it is unsuitable for application in scenarios where collaborative business encompasses numerous behavior paradigms.

## Time spent experiments

Regarding time performance measurements in our experiments, we conduct experiments and compare the performance of our method against the CM-SPADE and t-FCSM baseline methods, since the main time cost arises from mining frequent sequences. The experiments are conducted on a high-performance computer equipped with an Intel Core i9-10900K CPU (10 cores, 20 threads), 64GB RAM, and an NVIDIA GeForce RTX 3080 graphics card. This hardware configuration ensures that we could fairly evaluate the performance of various algorithms when processing large datasets.

The result of Time Spent Experiments is shown in Table 14. CM-SPADE and t-FCSM are general sequence pattern mining methods, not maximal sequence pattern mining methods. Therefore, after mining the general sequence patterns, a post-processing step is required to obtain the maximal sequence patterns. In our detailed analysis of the VMSP, CM-SPADE, and t-FCSM algorithms across different data scale scenarios, we identified key quantitative results that highlight the strengths and limitations of each algorithm. Initially, the VMSP algorithm demonstrated a training time 29% faster than CM-SPADE and 16% faster than t-FCSM in small data scale scenarios. In large data scale scenarios, VMSP's performance was even more impressive, with training times 17% faster than CM-SPADE and 12% faster than t-FCSM. These results clearly indicate VMSP's efficiency advantages.

Furthermore, we conducted an in-depth analysis of the algorithms' principles. VMSP's main advantages stem from its efficient data structures and optimized parallel computing strategies, which significantly reduce unnecessary data scans and computations when handling large datasets, thereby enhancing processing speed. In contrast, CM-SPADE, although faster in small-scale data processing, lacks memory usage efficiency and scalability when dealing

**Table 14. Results of time spent experiments.**

| Scenario | Method | Training Time (s) | Performance Improvement Comparison |
|---|---|---|---|
| Scenario #1 | Our Method | 48 | - |
| | CM-SPADE | 68 | Our Method training faster by 29% |
| | t-FCSM | 57 | Our Method training faster by 16% |
| Scenario #2 | Our Method | 145 | - |
| | CM-SPADE | 174 | Our Method training faster by 17% |
| | t-FCSM | 162 | Our Method training faster by 12% |

**Table 15. Parameters for impact analysis of support.**

| Scenario #1 | Scenario #2 | Expected Impact | Theoretical Basis |
|---|---|---|---|
| 0.07 | 0.008 | Low support level may lead to too many patterns and increased noise | Low threshold allows the identification of rarer behavior patterns, aiding exploratory analysis |
| 0.09 | 0.011 | Balance exploration and accuracy | Increasing support level helps reduce noise, but may overlook a few important patterns |
| 0.11 | 0.014 | Improve model generalization ability | Moderate support level can provide good pattern coverage while maintaining model interpretability |
| 0.13 | 0.017 | Reduces irrelevant patterns, enhancing clarity | High support focuses on prevalent behaviors, potentially missing less frequent yet relevant patterns |
| 0.15 | 0.02 | Risks overlooking critical rare patterns | Higher support tests the robustness of model under stringent conditions, ignoring sparse but significant patterns |
| 0.17 | 0.023 | May exclude important but infrequent patterns | Highest support settings challenge the model's ability to detect crucial anomalies under limited conditions |

with large datasets. The t-FCSM algorithm, while having specific advantages in time-series data analysis, exhibits weaker capabilities in processing complex datasets, particularly at larger scales, where efficiency and performance significantly decline.

These analytical findings reflect various factors to consider when selecting suitable algorithms, including data scale, algorithm efficiency, and specific requirements of application scenarios. They also illustrate how to choose the appropriate algorithm to optimize performance based on different business needs in practical applications. This thorough performance analysis and principled explanation help us better understand the applicability and potential application value of each algorithm.

## Impact of support

In this section, we analyze the impact of the support on mining behavior paradigm dependencies in our method.

The range of support values is shown in the Table 15, where support is an important parameter for measuring the frequency of behavior patterns. Here, the choice of support gradually increases from 0.07 to 0.17 for Scenario #1, and 0.008 to 0.023 for Scenario #2. The purpose is to observe whether the model can more accurately identify frequent behavior patterns as the support increases. Lower support allows less common behavior patterns to be considered important, while higher support emphasizes more common and consistent behavior patterns. Such settings help find the optimal support threshold to balance the model's generalization ability and the risk of overfitting.

As shown in the Tables 16 and 17, in Scenario #1, with the increase in support, precision and F1-Score initially rise and then fall, while recall remains at 100% at lower support levels. When the support is 0.11, Precision reaches its peak value of 94.37%, and F1-Score also achieves its highest value of 97.10%. As the support continues to increase, both Precision and

**Table 16. Experimental results of varying support of Scenario #1.**

| Support Degree | 0.07 | 0.09 | 0.11 | 0.13 | 0.15 | 0.17 |
|---|---|---|---|---|---|---|
| Precision (%) | 61.11 | 84.62 | 94.37 | 88 | 68.97 | 63.64 |
| Recall (%) | 100.00 | 100.00 | 100.00 | 100.00 | 90.91 | 73.64 |
| F1 (%) | 75.86 | 91.67 | 97.10 | 93.62 | 78.43 | 68.28 |

**Table 17. Experimental results of varying support of Scenario #2.**

| Support Degree | 0.008 | 0.011 | 0.014 | 0.017 | 0.02 | 0.023 |
|---|---|---|---|---|---|---|
| **Precision** (%) | 70.12 | 92.25 | 93.75 | 95.57 | 90.77 | 83.74 |
| **Recall** (%) | 100.00 | 100.00 | 100.00 | 100.00 | 93.71 | 90.88 |
| **F1** (%) | 82.44 | 95.97 | 96.77 | 97.73 | 92.22 | 87.16 |

F1-Score decline, possibly because higher support thresholds filter out fewer frequent patterns, which may include noise data, affecting the accuracy. At a support level of 0.13, Precision and F1-Score begin to drop significantly, although Recall remains high, indicating that at higher support levels, the model may miss some important patterns, leading to decreased Precision.

In Scenario #2, the impact of support on each metric is similar to that in Scenario #1. Precision and F1-Score reach their highest values at a support level of 0.017, at 95.57% and 97.73%, respectively. Compared to Scenario #1, Scenario #2 achieves higher Precision and F1-Score even at lower support levels (e.g., 0.008), indicating that in a more complex data environment, our method can effectively extract meaningful patterns at lower support levels. This phenomenon suggests that the dataset in Scenario #2 may contain more frequent patterns, allowing effective mining even at lower support thresholds.

To further intuitively observe the trend changes in the results, we visualize the results as shown in Figs 8 and 9. Similar experimental results in both scenarios show that as the support increases, the precision and F1 score exhibit an upward trend followed by a subsequent decline. This indicates that low support reduce the threshold of frequent sequence patterns, causing more noise sequences to be classified as frequent sequence patterns and introducing incorrect dependencies. Conversely, high support thresholds make it difficult to mine longer sequence patterns, resulting in the failure to filter out subsequences that belong to these longer

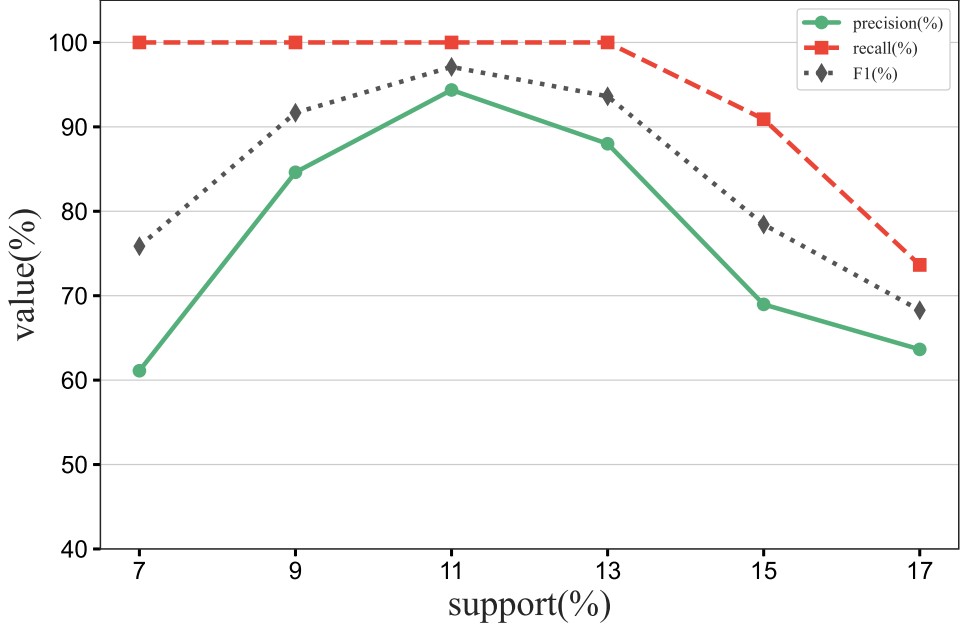

**Fig 8. Experimental results of varying support in Scenario #1.** The horizontal axis represents the magnitude of support, while the vertical axis represents the values of the indicators.

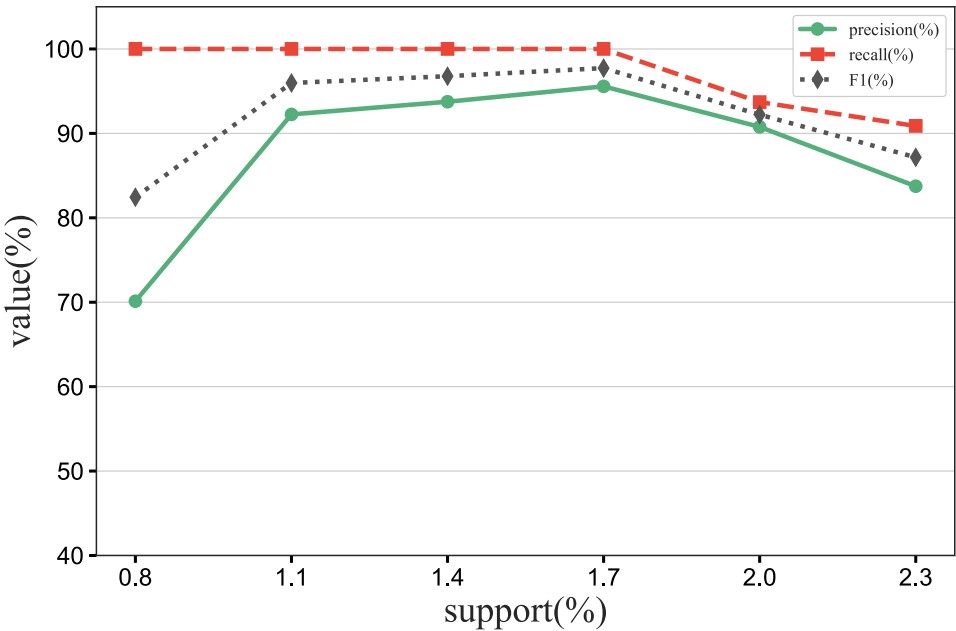

**Fig 9. Experimental results of varying support in Scenario #2.** The horizontal axis represents the magnitude of support, while the vertical axis represents the values of the indicators.

patterns, which may contain erroneous dependencies. Furthermore, appropriate supports ensure that all correct dependencies among behavior paradigms are mined. However, after the support exceeds a certain threshold, the recall starts to decline. The main reason may be that many frequent sequences cannot meet the new support and become infrequent sequences. These discarded sequences may contain some unique correct behavior paradigm dependencies, and the loss of these relationships leads to the decline of recall.

Overall, in both scenarios, Precision and F1-Score perform best at moderate support levels (0.11 for Scenario #1 and 0.017 for Scenario #2), while Recall maintains a high level across most support levels. This demonstrates that our mining method can effectively extract significant patterns in different scale scenarios, ensuring high accuracy and comprehensiveness. Particularly, in lower support situations, the performance in Scenario #2 significantly outperforms Scenario #1, further proving the advantage of our method in handling complex data environments.

## Impact of sliding window size

In this section, we analyze the impact of sliding window size on mining behavior paradigm dependencies in our method.

The size of the sliding window directly affects the range within which the model observes behavior patterns, which is shown in the Table 18. Smaller windows may not capture enough contextual information, leading to incomplete pattern recognition; larger windows may include too much irrelevant information, reducing the accuracy of recognition. By varying the window size from 5 to 10 for Scenario #1 and from 6 to 11 for Scenario #2, the experiment aims to assess the impact of different window sizes on model performance and find a balance point where the model can most effectively recognize and utilize the temporal dependencies in behavioral data. Besides, the support of the two scenarios is 11% and 1.7% respectively.

**Table 18. Parameters for impact analysis of sliding window size.**

| Scenario #1 | Scenario #2 | Expected Impact | Theoretical Basis |
|---|---|---|---|
| 5 | 6 | May miss important context, leading to incomplete patterns | Small windows lack sufficient context for accurate pattern recognition |
| 6 | 7 | Slightly better pattern capture but still may miss details | Slightly larger windows offer better context, enhancing pattern capture |
| 7 | 8 | Good balance of context and noise, high precision and recall | Moderate windows provide enough context without much noise |
| 8 | 9 | Better precision with more context but may include irrelevant data | Large windows improve pattern recognition with more context |
| 9 | 10 | Comprehensive context but reduced precision due to noise | larger windows capture more context but add noise |
| 10 | 11 | Highest recall but lowest precision due to excessive noise | Largest windows ensure no pattern is missed but reduce precision |

Experimental results are shown in the Tables 19 and 20. In Scenario #1, we observe that as the window size gradually increased, the model's precision significantly improved from 76% to 94.37%. This change indicates that the model can more accurately identify target entities or features in larger windows, reducing misidentification. Particularly when the window size reaches 8, the model achieves a precision of 94.37%, demonstrating that a larger contextual window provides the model with more information, enabling it to better understand and process complex patterns in the data. Concurrently, the recall rate reached 100% at a window size of 7 and maintains this level in subsequent tests. This result shows that when the window size reaches 7 or above, the model captures all relevant instances, missing no positive data. This is extremely important in practical applications where ensuring no loss of information is critical, especially in scenarios requiring highly accurate information retrieval. Additionally, the F1 score increases from 80.85% to 97.10%, this leap in performance further verifies that the model's overall effectiveness significantly enhances in larger windows. This improvement is not just an enhancement in a single metric but a result of combined improvements in precision and recall, clearly demonstrating the significant impact of window size on model performance.

In Scenario #2, the model's performance also shows a similar improvement trend. The precision gradually increases from 88.17% to 95.57%. Similar to Scenario #1, this growth reflects the model's enhanced capability to grasp data features in larger data windows, effectively distinguishing between positive and negative samples, thereby reducing misjudgments. The performance of the recall rate is also particularly noteworthy, reaching 100% at a window size of 8, indicating that the model can retrieve all relevant instances completely. Such a high recall rate is particularly crucial in many applications, such as in medical or legal document searches where missing critical information could lead to severe consequences. The increase in the F1

**Table 19. Experimental results of varying sliding window size for Scenario # 1.**

| Window size | 5 | 6 | 7 | 8 | 9 | 10 |
|---|---|---|---|---|---|---|
| Precision (%) | 76 | 84.62 | 88 | 94.37 | 91.67 | 88 |
| Recall (%) | 86.36 | 92.23 | 100 | 100 | 100 | 100 |
| F1 (%) | 80.85 | 88.26 | 93.62 | 97.10 | 95.65 | 93.62 |

**Table 20. Experimental results of varying sliding window size for Scenario # 2.**

| Window Size | 6 | 7 | 8 | 9 | 10 | 11 |
|---|---|---|---|---|---|---|
| Precision (%) | 88.17 | 91.25 | 95.57 | 92.49 | 89.49 | 85.4 |
| Recall (%) | 85.57 | 94.84 | 100 | 100 | 100 | 100 |
| F1 (%) F1 (%) | 86.85 | 93.01 | 97.73 | 96.10 | 94.45 | 92.13 |

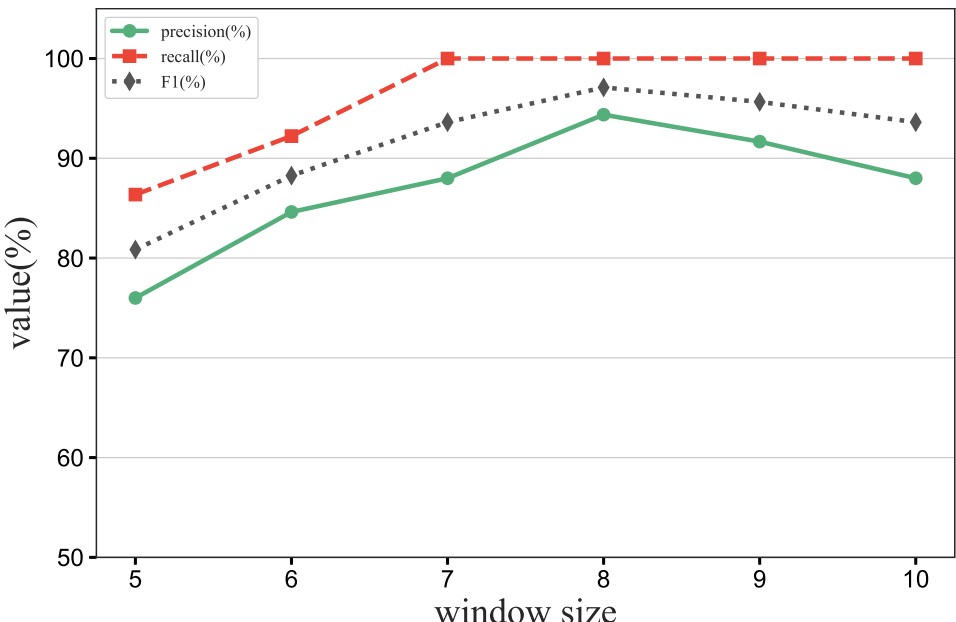

**Fig 10. Experimental results of varying sliding window size in Scenario #1.** The horizontal axis represents the size of the sliding window, while the vertical axis represents the values of the indicators.

score from 86.85% to 97.73% further certifies that with larger window settings, the overall performance of the model is significantly boosted. This enhancement is due to the synchronous improvements in precision and recall, ensuring the model's robustness and reliability when handling complex and variable data.

Visualization results of the two scenarios are shown in Figs 10 and 11. In both scenarios, changing the sliding window size while keeping the support constant yields a similar effect. Precision and F1 initially increase then decrease while recall decreases gradually as the sliding window size decreases below a certain threshold. The precision decreases when the window size reduces, possibly because a smaller window can only mine shorter sequence patterns, i.e., subsequences contained in longer sequences cannot be filtered out, and these short sequences may contain incorrect behavior paradigm dependencies. The decrease in precision as the window size increases may be because larger windows introduce more noise sequence fragments, thereby mining incorrect behavior paradigm dependencies. Additionally, the main reason for the reduction of recall caused by the reduction of window size may be that the smaller window causes the behavior paradigms belonging to the same long collaborative business to be divided into different sequences, resulting in the loss of the dependencies between these behavior paradigms.

Through detailed analysis of Scenarios #1 and #2, it is clear that window size plays a key role in enhancing model performance, and the proposed method demonstrates good adaptability and effectiveness across different types of data scenarios.

## Impact of data contribution in data pricing

Data contribution, as a data quality dimension based on data behavior, can impact the calculation of data costs from the perspective of data behavior, thus influencing data pricing. In order to analyze the impact of introducing data contribution on data pricing, we choose the following two cost calculation methods for comparison.

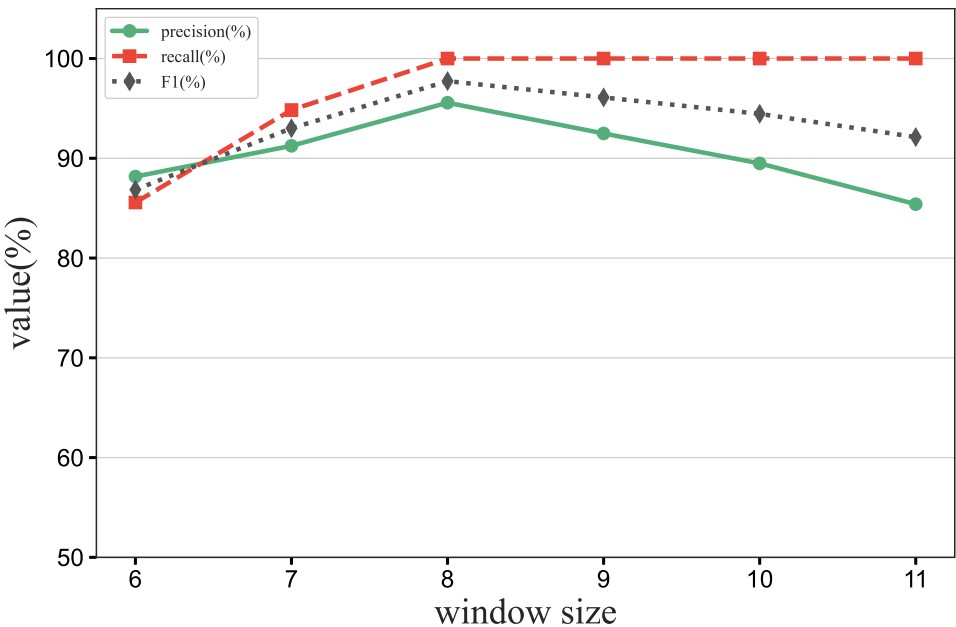

**Fig 11. Experimental results of varying sliding window size in Scenario #2.** The horizontal axis represents the size of the sliding window, while the vertical axis represents the values of the indicators.

**Comparison methods.**

- **Baseline**
  Following the noisy linear query settings in [71, 77], data costs are solely evaluated in terms of privacy compensation. Specifically, in round $t$, a privacy leakage mechanism based on differential privacy and a tanh-based privacy compensation function are employed to generate the privacy compensation of the data provider. Subsequently, the privacy compensation is used to generate a $k$-dimensional feature vector $\boldsymbol{h_t}$, which is then L2-normalized, i.e., $\|L2(\boldsymbol{h_t})\|_2 = 1$. The $cost_t$ is the sum of the components of $L2(\boldsymbol{h_t})$, i.e., $cost_t = \sum_i L2(\boldsymbol{h_t})_i$, $i = 1, 2 \ldots$, $k$. The pricing algorithm employs the ellipsoid-based data pricing (**EBDP**) mechanism [77].

- **Privacy-based Pricing Incorporating Data Contribution (PPIC)**
  To ensure that the proportions of data contribution and privacy compensation in the cost are roughly equal, we consider adding the contribution of the current trading data's database to each component of the L2-normalized privacy compensation. Furthermore, for convergence, the data cost needs to be L2-normalized overall [77], i.e., $cost_t = \sum_i L2(L2(\boldsymbol{h_t}) + \alpha * x_{provider} * \mathbf{1})_i$, $i = 1, 2, \ldots, k$, where $\alpha$ represents the weight to adjust the proportion between data contribution and privacy compensation (chosen as 0.1, 0.5, 1.0, 1.5, and 2.0 in the experiments, denoted as PPIC_alpha0.1, PPIC_alpha0.5, PPIC_alpha1.0, PPIC_alpha1.5, and PPIC_alpha2.0 respectively), $x_{provider}$ stands for the contribution of the database storing trading data, and $\mathbf{1}$ is a $k$-dimensional vector with all components equal to 1.

To evaluate the pricing performance, we simulate 100,000 trading rounds and utilize cumulative regret and regret ratio as evaluation metrics, as detailed in Table 21.

**Parameter settings.** As shown in Table 22, in the impact of data contribution in data pricing, the hyperparameter $\alpha$ is used to adjust the proportion between data contribution and privacy compensation in the data pricing model. Specifically, the value of $\alpha$ determines the

**Table 21. Evaluation indicators in data pricing.**

| Metrics | Definition |
|---------|-----------|
| Cumulative regret | Sum of regrets in the first T rounds of transactions $\sum_{t=1}^{T} R_t$ |
| Regret ratio | The ratio of cumulative regret and cumulative market value of the first T rounds $\sum_{t=1}^{T} R_t / \sum_{t=1}^{T} v_t$ |

**Table 22. Parameter specific values.**

| Parameter | Values | Description |
|-----------|--------|-------------|
| $\alpha$ | 0.1, 0.5, 1.0, 1.5, 2.0 | The weight adjusting the proportion between data contribution and privacy compensation. |
| Rounds (Cumulative Regret Experiment) | 100, 20000, 40000, 60000, 80000, 100000 | Total number of simulated data trading rounds for cumulative regret experiments. |
| Rounds (Regret Ratio Experiment) | 1, 10, 100, 1000, 10000, 100000 | Total number of simulated data trading rounds for regret ratio experiments. |

relative weight of data contribution in the total cost of each transaction. Experimental results show that increasing the $\alpha$ value leads to a greater impact of data contribution on the pricing strategy, thereby reducing cumulative regret and regret ratio. This indicates that incorporating data contribution into data pricing can effectively optimize the pricing strategy.

The simulation rounds (Rounds) are divided into two categories based on the type of experiment: cumulative regret experiments and regret ratio experiments. For cumulative regret experiments, the number of rounds ranges from 100 to 100000, aimed at evaluating the pricing mechanism's performance over varying transaction frequencies. For regret ratio experiments, the rounds range from 1 to 100000, focusing on assessing the efficiency of the pricing strategy at different stages of the simulation. These settings ensure a comprehensive analysis of the data pricing mechanism under various conditions, validating the effectiveness and stability of the proposed model.

**Experimental results.**   In this section, we analyze the impact of introducing data contribution in data pricing under the two scale scenarios. The experimental results are shown in Figs 12–15.

These four tables provides crucial data comparing the effectiveness of different methods (EBDP and PPIC with varying alpha values) over multiple rounds in two different scenarios. The cumulative regret tables (Tables 23 and 24) give a total measure of regret over time, while the regret ratio tables (Tables 25 and 26) provide a comparative measure of performance efficiency at different points during the experiment. This data is vital for understanding the performance and efficiency of each method in the given scenarios.

We can observe from the results of the two scenarios that as the number of trading round $t$ increases, the cumulative regret gradually rises and then stabilizes. This is because data providers need to publish exploration prices and update the range of $\theta$ in more transactions, making the posted price converge towards the true market value of the data. Meanwhile, the regret ratio exhibits an overall decreasing trend, which can be attributed to the fact that as the trading round $t$ increases, the gap between $p_t$ and the $v_t$ gradually diminishes, resulting in a slower increase in cumulative regret compared to cumulative market value. It is worth noting that around 100 trading rounds, regret ratio shows a short-term increase trend, which is due to the intense changes in the pricing mechanism when selecting exploration and exploitation prices.

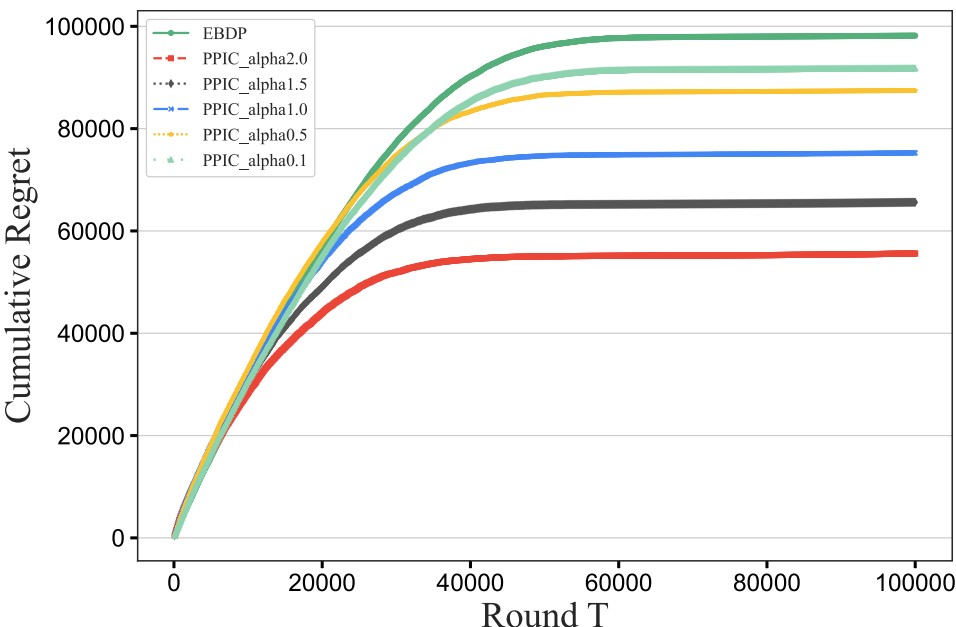

**Fig 12. Comparisons of cumulative regret in Scenarios #1.**

In Scenario #1, PPIC significantly reduces cumulative regret compared to the baseline. Furthermore, as the weight of data contribution increases, cumulative regret experiences a decrease. This is primarily attributed to the additional cost information introduced by data contribution, facilitating the update of the range of $\theta$ during the pricing process. It narrows the gap between the posted price and the market value, thereby reducing regret. As $\alpha$ increases, the proportion of data contribution in the cost gradually rises, and with the increase of $t$, data

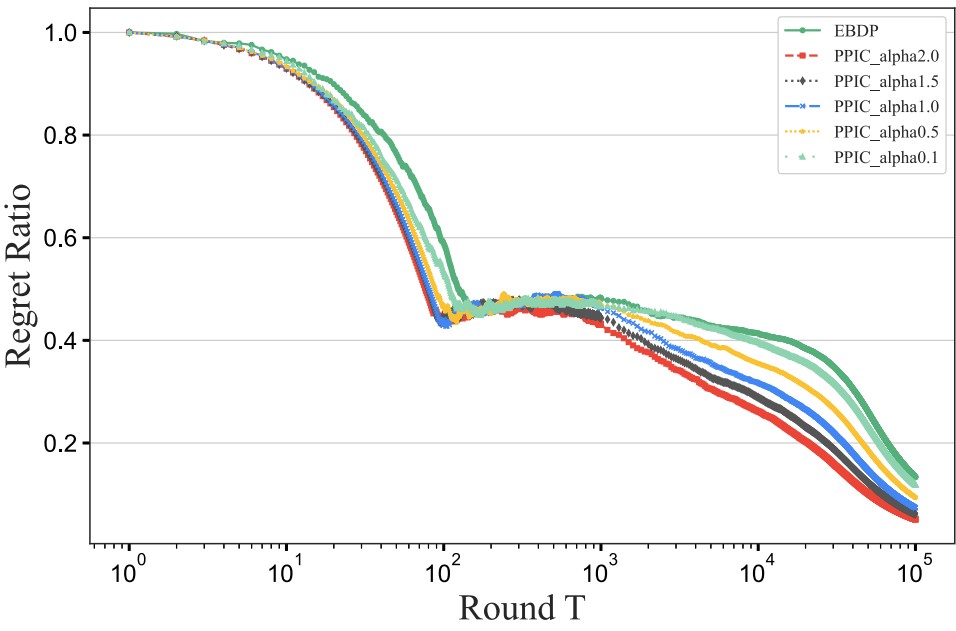

**Fig 13. Comparisons of regret rate in Scenarios #1.**

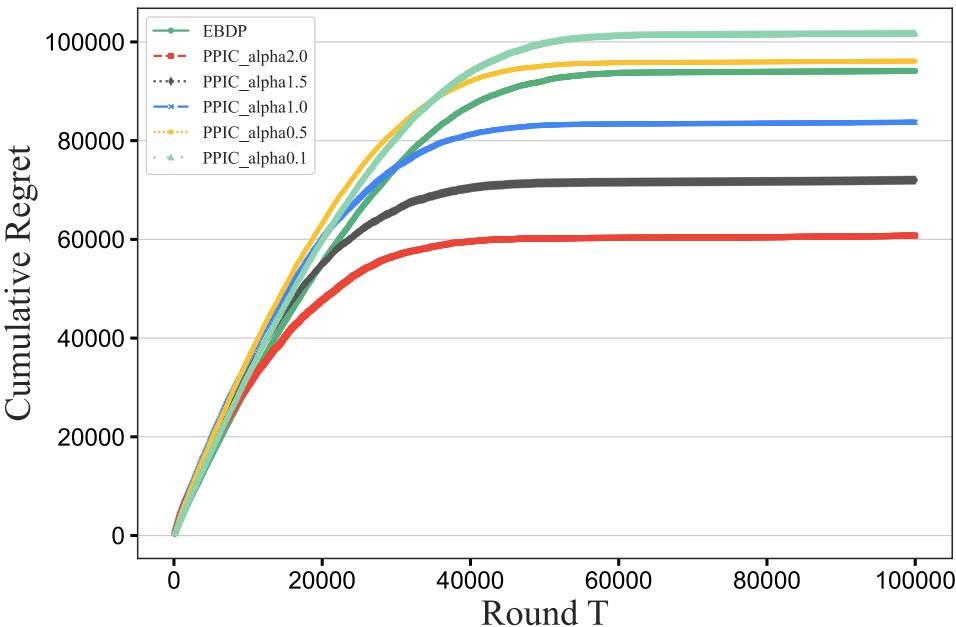

**Fig 14. Comparisons of cumulative regret in Scenarios #2.**

providers are more inclined to determine the price based on the data contribution, aiming to achieve maximum returns. When the contribution weight $\alpha$ is set to 1.0, 1.5, and 2.0 respectively, the cumulative regret of PPIC is reduced by 10.93%, 23.34%, and 43.38% compared to the baseline. Additionally, with the increase in the contribution weight, the final regret ratio continues to decrease. Notably, in the baseline, the final regret ratio is 0.1336, while PPIC with

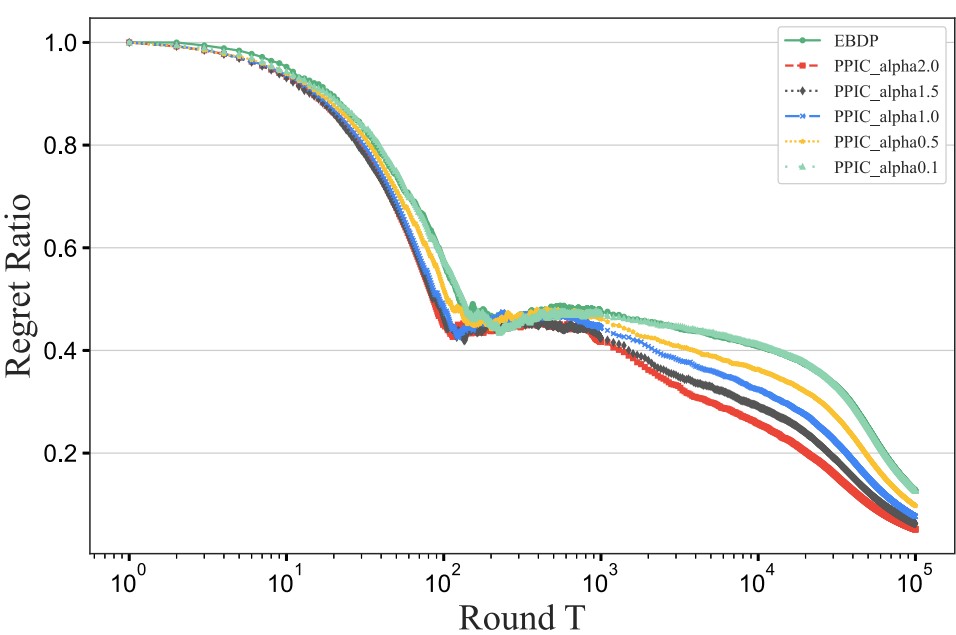

**Fig 15. Comparisons of regret rate in Scenarios #2.**

**Table 23. Cumulative regret of Scenario #1.**

| Rounds | EBDP | PPIC_alpha2.0 | PPIC_alpha1.5 | PPIC_alpha1.0 | PPIC_alpha0.5 | PPIC_alpha0.1 |
|---|---|---|---|---|---|---|
| 100 | 437.654493 | 476.0199345 | 477.1021872 | 448.3982526 | 434.0220308 | 400.6988604 |
| 20000 | 56625.84285 | 43952.16123 | 49029.17673 | 54082.01185 | 57761.03243 | 55166.53524 |
| 40000 | 90222.05005 | 54498.21942 | 64235.68581 | 73317.637 | 83381.42926 | 85412.40259 |
| 60000 | 97704.49584 | 55133.74991 | 65205.5126 | 74900.54751 | 87093.66132 | 91419.55962 |
| 80000 | 97997.13743 | 55282.73816 | 65350.4221 | 75047.28918 | 87263.65065 | 91621.90396 |
| 100000 | 98170.75967 | 55583.12662 | 65611.15176 | 75260.97367 | 87440.38349 | 91797.52987 |

**Table 24. Cumulative regret of Scenario #2.**

| Rounds | EBDP | PPIC_alpha2.0 | PPIC_alpha1.5 | PPIC_alpha1.0 | PPIC_alpha0.5 | PPIC_alpha0.1 |
|---|---|---|---|---|---|---|
| 100 | 419.9544357 | 533.0817138 | 524.8127 | 531.1317038 | 518.9786861 | 466.0662456 |
| 20000 | 55176.50771 | 47665.24163 | 55020.1112 | 60195.72569 | 63157.89619 | 60104.37019 |
| 40000 | 87046.3129 | 59562.14087 | 70396.5323 | 81270.9539 | 92054.2932 | 93994.1309 |
| 60000 | 93733.45263 | 60300.67561 | 71582.56382 | 83382.15357 | 95773.9288 | 101267.1433 |
| 80000 | 93945.30333 | 60450.95864 | 71742.75382 | 83530.85647 | 95930.86459 | 101571.3489 |
| 100000 | 94120.51584 | 60748.69778 | 72000.88182 | 83742.47978 | 96113.91057 | 101745.0805 |

**Table 25. Regret ratio of Scenario #1.**

| Rounds | EBDP | PPIC_alpha2.0 | PPIC_alpha1.5 | PPIC_alpha1.0 | PPIC_alpha0.5 | PPIC_alpha0.1 |
|---|---|---|---|---|---|---|
| 1 | 1 | 1 | 1 | 1 | 1 | 1 |
| 10 | 0.947835633 | 0.929227801 | 0.929882903 | 0.930218355 | 0.93308081 | 0.944230379 |
| 100 | 0.588540249 | 0.440662112 | 0.45009212 | 0.440945556 | 0.465907601 | 0.532739305 |
| 1000 | 0.480906527 | 0.430454679 | 0.444265307 | 0.463236616 | 0.467521268 | 0.467685337 |
| 10000 | 0.413353555 | 0.261447166 | 0.289245158 | 0.31682957 | 0.354982832 | 0.396373646 |
| 100000 | 0.133564527 | 0.051323045 | 0.06188688 | 0.073909182 | 0.093871831 | 0.119041503 |

**Table 26. Regret ratio of Scenario #2.**

| Rounds | EBDP | PPIC_alpha2.0 | PPIC_alpha1.5 | PPIC_alpha1.0 | PPIC_alpha0.5 | PPIC_alpha0.1 |
|---|---|---|---|---|---|---|
| 1 | 1 | 1 | 1 | 1 | 1 | 1 |
| 10 | 0.952533615 | 0.932918057 | 0.930909716 | 0.93662794 | 0.939273777 | 0.943257893 |
| 100 | 0.573446381 | 0.454946021 | 0.460763044 | 0.484468785 | 0.519124762 | 0.574535524 |
| 1000 | 0.471825081 | 0.418068224 | 0.424258663 | 0.445295125 | 0.46208708 | 0.471490635 |
| 10000 | 0.408054976 | 0.25740248 | 0.290781519 | 0.323186083 | 0.363036349 | 0.411791211 |
| 100000 | 0.127275625 | 0.051574501 | 0.062763259 | 0.076522314 | 0.097346079 | 0.126900118 |

a contribution weight $\alpha$ of 2.0 achieves a final regret ratio of 0.0513, representing a 61.8% reduction. Despite data providers having a good estimation of the data value after a sufficient number of trading rounds, the impact of contribution on the posted price remains significant, demonstrating the effectiveness of incorporating data contribution.

The conclusions from Scenario #2 are similar to those in Scenario #1. Based on the experiment results related to cumulative regret, when the contribution weight $\alpha$ is set to 1.0, 1.5, and

2.0 respectively, the cumulative regret of PPIC is reduced by 11.03%, 23.50%, and 35.46% compared to the baseline. Regarding the experiment results related to the regret ratio, the final regret ratio of baseline is 0.1273, whereas, with the $\alpha$ of 2.0, the final regret ratio is 0.0516, representing a reduction of 59.8%.

## Limitation and discussion

This study acknowledges several limitations that warrant further investigation. While our method for assessing data contribution in decentralized sharing scenarios has shown promising results, the reliance on specific blockchain architectures may affect the generalizability of our findings. Additionally, the computational intensity of maximal sequential pattern mining could pose challenges in larger, more dynamic datasets.

## Limitation

- **Sensitivity to Hyperparameters**
  Our method relies on multiple critical hyperparameter settings, such as the support threshold for frequent sequence mining and the window size for behavioral pattern extraction. These hyperparameters may require different configurations across various datasets or application contexts to ensure effectiveness and accuracy. A primary issue with hyperparameter settings is their high sensitivity; different parameter values can significantly impact the final assessment outcomes. For instance, varying the support threshold from 5% to 15% led to a 20% decrease in recall, as valuable patterns below the new threshold were excluded. Similarly, the setting of the window size needs to balance capturing sufficient behavioral information and avoiding redundant data. Optimizing these hyperparameters typically necessitates extensive experimentation and tuning, which is not only time-consuming but may also not guarantee a globally optimal solution. Additionally, varying data characteristics and application demands might lead to significant discrepancies in optimal hyperparameter settings, increasing complexity and uncertainty in practical applications.

- **Dependency on Sequence Length**
  Our approach shows a high dependency on the length of sequences used in mining behavioral patterns. Short sequences might not adequately capture contextual information about user behaviors or data exchanges, leading to inaccuracies in the evaluation results. Conversely, long sequences may include irrelevant information and noise, increasing the complexity of data processing and computational resource consumption. The choice of sequence length substantially affects the algorithm's performance and the accuracy of the results. For shorter sequences, our method might require the integration of more contextual information or additional data features to compensate for insufficient data. In experiments, reducing window size from 10 to 5 time units caused a drop in F1 score from 0.75 to 0.55, reflecting the loss of contextual continuity. For longer sequences, it becomes crucial to develop more efficient data processing and compression techniques to mitigate computational loads and enhance processing speeds. Moreover, different application scenarios and data types might have varying requirements for sequence length, demanding that our method adapt flexibly in real-world applications.

- **Computational Complexity and Resource Demands**
  The computational complexity and resource requirements of our method escalate when dealing with large-scale datasets. The processes of mining maximal sequence patterns and constructing behavioral paradigm graphs require substantial computational resources and memory, which could become a bottleneck when handling vast amounts of data. As data

scales increase, the algorithm's time and space complexities grow exponentially, potentially leading to prolonged processing times and memory exhaustion. During the evaluation of a dataset comprising over 10 million transactions, computational time increased tenfold as the volume of data doubled, underscoring the scalability challenges. To address these issues, we need to develop more efficient algorithms and data structures to optimize computation, reducing unnecessary computational steps. Furthermore, leveraging distributed computing and parallel processing techniques can significantly enhance the scalability and processing capabilities of our method. However, this also implies that sufficient computational resources and technical support are essential in practical applications. Thus, reducing computational complexity and resource demands while ensuring algorithmic accuracy remains a critical research direction for us.

## Discussion

Despite our method's demonstrated efficacy and accuracy in experimental settings, the aforementioned limitations indicate several areas for improvement and optimization. To overcome these limitations, we plan to conduct in-depth research in the following areas.

- **Optimizing Hyperparameter Tuning Methods**
  We will explore automated hyperparameter tuning techniques, such as those based on Bayesian optimization or evolutionary algorithms, to reduce manual tuning efforts and enhance tuning efficiency.

- **Improving Sequence Processing Techniques**
  For sequences of varying lengths, we aim to develop more adaptive sequence processing and compression technologies to ensure that critical behavioral information is captured while minimizing noise and redundancy.

- **Enhancing Computational Efficiency and Scalability**
  By introducing more efficient algorithms and data structures, combined with distributed computing and parallel processing, we plan to boost the processing power and scalability of our method, ensuring it remains effective even on large-scale datasets. Through these improvements, we believe we can further enhance the practical utility and application value of our method, providing more effective and reliable solutions for assessing data contribution in decentralized sharing and exchange scenarios.

## Conclusion

This work focuses on the assessment of data contribution in decentralized sharing and exchange consortium blockchains. Utilizing metadata sequences of data sharing and exchange behaviors recorded on the consortium blockchain, we propose a data contribution assessment method based on the maximal sequential patterns of sharing and exchange behavior paradigms. This method aims to explore dependencies among sharing and exchange behavior paradigms and determine the weights of these paradigms. The maximal sequential pattern mining algorithm helps filter out redundant or potentially erroneous behavior paradigm dependencies, enhancing the accuracy of the discovered dependencies and their weight. Finally, combining the data volume of data behaviors allows for the calculation of the contribution of each participating database in the consortium blockchain. The effectiveness of the proposed method and the positive role of data contribution in data pricing are demonstrated through

experiments conducted in two different scale scenarios. pecifically, compared to the most competitive baseline, the improvements of mean average precision in two scenarios are 6% and 8%.

## Future work

This article introduces and defines a new context quality metric, data contribution, and explores methods to assess this metric in scenarios of data sharing and exchange. We have proposed a method for assessing data contribution based on sharing and exchange behavior paradigms and maximal sequence patterns. This method has been empirically validated on open-source systems and simulated scenarios, demonstrating its capability to address data contribution assessment issues in respective contexts to some extent. However, there remain several areas for improvement and refinement in our study, and future work could continue exploring the following aspects.

- **Improvements in Data Contribution Assessment Methods**.
  Our current method only considers direct dependencies in mining relationships. Future work could incorporate indirect and transitive dependencies to more accurately mine the dependencies and their strengths between different behavioral paradigms. The research primarily focuses on data sharing scenarios driven by collaborative business between different systems, considering only temporary sharing of data usage rights and multiparty data behavior types. Further improvements should integrate characteristics of non-collaborative business-driven sharing scenarios, such as data trading and data services, to extend the applicability of the current method.

- **Introduction of Adaptive Hyperparameter Optimization Techniques**.
  To address the deficiencies in hyperparameter settings of the current method, future work could involve adaptive hyperparameter optimization techniques, such as Bayesian optimization or genetic algorithms, to automatically adjust key parameters like support thresholds, enhancing the method's applicability and robustness.

- **Reducing Dependency on Sequence Length**.
  In order to reduce the dependency on the length of data sequences, future approaches could explore combining sequence compression techniques or graph-based pattern mining methods to enhance the efficiency and accuracy when handling long sequences, adapting to a more diverse range of data scenarios. Through these improvements and extensions, the method proposed in this article is expected to play a greater role in a broader range of practical applications, providing a more comprehensive and precise solution for assessing data contribution.

## Author Contributions

**Conceptualization:** Wenjun Ke, Yulin Liu.

**Data curation:** Yikai Guo.

**Formal analysis:** Yikai Guo, Rui Wang, Peng Wang.

**Investigation:** Jiahao Wang, Yikai Guo, Rui Wang.

**Methodology:** Wenjun Ke, Yulin Liu.

**Project administration:** Rui Wang, Peng Wang.

**Software:** Yulin Liu, Zangbo Chi, Rui Wang.

**Supervision:** Zhi Fang, Yikai Guo.

**Validation:** Jiahao Wang, Zhi Fang, Zangbo Chi.

**Visualization:** Jiahao Wang, Zangbo Chi, Peng Wang.

**Writing – original draft:** Wenjun Ke, Yulin Liu.

**Writing – review & editing:** Jiahao Wang, Zhi Fang, Zangbo Chi, Peng Wang.

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
