## [Decision Letter · Decision Letter 0]

30 Apr 2024

PONE-D-24-06576DecentralDC: Assessing Data Contribution under Decentralized Sharing and Exchanging BlockChainPLOS ONE

Dear Dr. Liu,

Thank you for submitting your manuscript to PLOS ONE. After careful consideration, we feel that it has merit but does not fully meet PLOS ONE’s publication criteria as it currently stands. Therefore, we invite you to submit a revised version of the manuscript that addresses the points raised during the review process. Please submit your revised manuscript by Jun 14 2024 11:59PM. If you will need more time than this to complete your revisions, please reply to this message or contact the journal office at plosone@plos.org. Please include the following items when submitting your revised manuscript:A rebuttal letter that responds to each point raised by the academic editor and reviewer(s). You should upload this letter as a separate file labeled 'Response to Reviewers'.A marked-up copy of your manuscript that highlights changes made to the original version. You should upload this as a separate file labeled 'Revised Manuscript with Track Changes'.An unmarked version of your revised paper without tracked changes. You should upload this as a separate file labeled 'Manuscript'.

We look forward to receiving your revised manuscript.

Kind regards,

Emanuele Crisostomi, PhD

Academic Editor

PLOS ONE

Journal Requirements:

"This work was supported by National Science Foundation of China (Grant Nos.62376057) and the Start-up Research Fund of Southeast University (RF1028623234)."

"This work was supported by National Science Foundation of China (Grant

Nos.62376057) and the Start-up Research Fund of Southeast University

(RF1028623234)."

"This work was supported by National Science Foundation of China (Grant Nos.62376057) and the Start-up Research Fund of Southeast University (RF1028623234)."

**Additional Editor Comments:**

Both reviewers recommend that the manuscript should be revised before it can be accepted for publication, and provide a list of suggestions for its improvement. It is thus recommended that the authors carefully follow the list of suggestions to prepare the revised manuscript. 

Reviewers' comments:

Reviewer's Responses to Questions

**Comments to the Author**

1. Is the manuscript technically sound, and do the data support the conclusions?

Reviewer #1: Yes

Reviewer #2: Partly

2. Has the statistical analysis been performed appropriately and rigorously? 

Reviewer #1: Yes

Reviewer #2: Yes

3. Have the authors made all data underlying the findings in their manuscript fully available?

Reviewer #1: Yes

Reviewer #2: Yes

4. Is the manuscript presented in an intelligible fashion and written in standard English?

Reviewer #1: Yes

Reviewer #2: Yes

5. Review Comments to the Author

Reviewer #1: 1. Describe dataset features in more details and its total size and size of (train/test).

2. Pseudocode and algorithm steps need to be inserted.

3. Time spent need to be measured in the experimental results.

4. Limitation and Discussion Sections need to be inserted.

5. All metrics need to be calculated in the experimental results.

6. The parameters used for the analysis must be provided in table

7. The architecture of the proposed model must be provided

8. Address the accuracy/improvement percentages in the abstract and in the conclusion sections, as well as the significance of these results.

9. The authors need to make a clear proofread to avoid grammatical mistakes and typo errors.

10. Add future work in last section (conclusion) (if any)

11. To improve the Related Work and Introduction sections authors are recommended to review this highly related research work paper:

A) Unlocking the power of blockchain in education: An overview of innovations and outcomes

B) Topic Extraction and Interactive Knowledge Graphs for Learning Resources

C) Privacy issues of public Wi-Fi networks

Reviewer #2: Here are some comments:

1- The abstract of the paper is not concise and redundant, which needs to be further reduced.

2- How to verify the protocol proposed in the paper? Is there a specific application system?

3- The “Introduction” and “Related work” sections lack of enough references. I strongly recommend that the author improve this section by adding references that support all the claims and motivation of the problem. The author may precisely and comprehensively point out the current issues and existing solutions. I suggest adding more related reference.

4-In related works, the limitations of each work must be mentioned, and a summary at the end explains how the proposed work was able to overcome them.

5- The algorithms mentioned in this paper need to be compared with the current popular algorithms, and analyze their advantages in theory, the depth of the comparison method is not enough.

6- The theoretical performance verification analysis of the main methods in this paper is insufficient.

7- As an academic research, there is a serious lack of experimental proof, verification and comparison of various schemes.

6. PLOS authors have the option to publish the peer review history of their article (what does this mean?). If published, this will include your full peer review and any attached files.

Reviewer #1: **Yes: **Tarek Abd El-Hafeez

Reviewer #2: No

---

## [Author Response · Author response to Decision Letter 0]

18 Jul 2024

Dear Editors and Reviewers, 

 We appreciate your letter and reviewers' constructive comments on our manuscript entitled DecentralDC: Assessing Data Contribution under Decentralized Sharing and Exchanging BlockChain." We have carefully studied all comments and revised the manuscript following the suggestions. These comments are valuable and helpful for revising and improving our manuscript. The updated contents are highlighted in yellow in the revised manuscript, and the responses to the comments are detailed in "Response to Letters.pdf," which has been uploaded in the "Attach Files" due to space limitations here . Once again, we thank your reviews for possible publication. 

Best regards,

The authors

---

## [Decision Letter · Decision Letter 1]

6 Sep 2024

DecentralDC: Assessing Data Contribution under Decentralized Sharing and Exchanging BlockChain

PONE-D-24-06576R1

Dear Dr. Liu,

We’re pleased to inform you that your manuscript has been judged scientifically suitable for publication and will be formally accepted for publication once it meets all outstanding technical requirements.

Kind regards,

Emanuele Crisostomi, PhD

Academic Editor

PLOS ONE

Additional Editor Comments (optional):

Comments from PLOS Editorial Office: We note that one or more reviewers has recommended that you cite specific previously published works in an earlier round of revision. As always, we recommend that you please review and evaluate the requested works to determine whether they are relevant and should be cited. It is not a requirement to cite these works and you may remove them before the manuscript proceeds to publication. We appreciate your attention to this request.

Reviewers' comments:

Reviewer's Responses to Questions

**Comments to the Author**

1. If the authors have adequately addressed your comments raised in a previous round of review and you feel that this manuscript is now acceptable for publication, you may indicate that here to bypass the “Comments to the Author” section, enter your conflict of interest statement in the “Confidential to Editor” section, and submit your "Accept" recommendation.

Reviewer #1: All comments have been addressed

Reviewer #3: All comments have been addressed

2. Is the manuscript technically sound, and do the data support the conclusions?

Reviewer #1: (No Response)

Reviewer #3: Yes

3. Has the statistical analysis been performed appropriately and rigorously? 

Reviewer #1: (No Response)

Reviewer #3: Yes

4. Have the authors made all data underlying the findings in their manuscript fully available?

Reviewer #1: (No Response)

Reviewer #3: Yes

5. Is the manuscript presented in an intelligible fashion and written in standard English?

Reviewer #1: (No Response)

Reviewer #3: Yes

6. Review Comments to the Author

Reviewer #1: The updated manuscript, which addresses previous comments and suggestions, has been evaluated positively. The revised submission demonstrates significant improvement and provides valuable insights relevant to the research community. I recommend accepting it for publication.

Reviewer #3: All comments and concerns raised in the previous review round have been resolved. The authors have made significant improvements to the manuscript, addressing all the feedback provided.

7. PLOS authors have the option to publish the peer review history of their article (what does this mean?). If published, this will include your full peer review and any attached files.

Reviewer #1: No

Reviewer #3: No

---

## [Editor Report · Acceptance letter]

11 Oct 2024

PONE-D-24-06576R1 

PLOS ONE

Dear Dr. Liu, 

I'm pleased to inform you that your manuscript has been deemed suitable for publication in PLOS ONE. Congratulations! Your manuscript is now being handed over to our production team.

Kind regards, 

on behalf of

Professor Emanuele Crisostomi 

Academic Editor

PLOS ONE